# Purging of highly deleterious mutations through severe bottlenecks in Alpine ibex

Christine Grossen [1✉], Frédéric Guillaume[1], Lukas F. Keller [1,2] & Daniel Croll [3✉]

Human activity has caused dramatic population declines in many wild species. The resulting bottlenecks have a profound impact on the genetic makeup of a species with unknown consequences for health. A key genetic factor for species survival is the evolution of deleterious mutation load, but how bottleneck strength and mutation load interact lacks empirical evidence. We analyze 60 complete genomes of six ibex species and the domestic goat. We show that historic bottlenecks rather than the current conservation status predict levels of genome-wide variation. By analyzing the exceptionally well-characterized population bottlenecks of the once nearly extinct Alpine ibex, we find genomic evidence of concurrent purging of highly deleterious mutations but accumulation of mildly deleterious mutations. This suggests that recolonization bottlenecks induced both relaxed selection and purging, thus reshaping the landscape of deleterious mutation load. Our findings highlight that even populations of ~1000 individuals can accumulate mildly deleterious mutations. Conservation efforts should focus on preventing population declines below such levels to ensure long-term survival of species.

[1] Department of Evolutionary Biology and Environmental Studies, University of Zurich, CH-8057 Zurich, Switzerland. [2] Zoological Museum, University of Zurich, Karl-Schmid-Strasse 4, CH-8006 Zurich, Switzerland. [3] Laboratory of Evolutionary Genetics, Institute of Biology, University of Neuchâtel, CH-2000 Neuchâtel, Switzerland. ✉email: christine.grossen@uzh.ch; daniel.croll@unine.ch

Climate change and pressure from human activities such as hunting caused profound changes in the population and demographic structure of many species[1]. This is because extinction events and subsequent recolonization severely alter the genetic makeup[2]. The demographic changes have important consequences for wildlife management and the conservation of endangered species[2] including raising the risk of genetic disorders[3–9]. However, nearly all plant and animal populations including humans suffered from temporary reductions in population size—so-called bottlenecks. Bottlenecks increase genetic drift and inbreeding, which leads to a loss of genetic variation, reduces the efficacy of natural selection, and increases the expression of deleterious recessive mutations[10–12]. The expression of recessive mutations under inbreeding creates the potential for selection to act against these mutations. This process known as purging reduces the frequency of deleterious mutations depending on the degree of dominance and the magnitude of the deleterious effects[13]. Because purging depends on levels of inbreeding, bottlenecks tend to purge highly deleterious, recessive mutations unless population sizes are extremely low[13–15]. Bottlenecks also increase genetic drift and reduce the efficacy of selection[16]. This allows mildly deleterious mutations to drift to substantially higher frequencies[4,6,17]. Hence, bottlenecks generate complex dynamics of deleterious mutation frequencies due to the independent effects of purging and reduced selection efficacy[13–15,18,19].

A major gap in our understanding is how reduced selection efficacy and purging jointly determine the mutation load in wild populations. Theoretical predictions are well established[13,15,18,20] but empirical evidence is conflicting[7,9,21] including for humans (see e.g[22–30].). Previous research used changes in fitness to infer possible purging events[7,20,31–33], but changes in fitness can result from causes unrelated to purging[12,34]. Direct evidence for purging exists only for isolated mountain gorilla populations that split off larger lowland populations ~20,000 years ago[35]. However, it remains unknown how recent, dramatic bottleneck events on the scale caused by human activity impacts levels of deleterious mutations in the wild. Here, we take advantage of exceptionally well characterized repeated bottlenecks during the reintroduction of the once near-extinct Alpine ibex to retrace the fate of deleterious mutations. Alpine ibex were reduced to ~100 individuals in the 19th century in a single population in the Gran Paradiso region of Northern Italy[36]. In less than a century, a census size of ca. 50,000 individuals has been re-established across the Alps. Thus, the population bottleneck of Alpine ibex is among the most dramatic recorded for any successfully restored species. The recolonization efforts focused on founding local populations across distinct mountain ranges with very limited opportunities for gene flow. Some successfully established populations were used to initiate secondary or tertiary populations elsewhere. As a consequence, most extant populations experienced two to four, well-recorded bottlenecks leaving strong footprints of low genetic diversity[37,38].

We find exceptionally low genome-wide variation and an accumulation of deleterious mutations in Alpine ibex compared to most closely related species. Over the course of population reintroductions, Alpine ibex populations that experienced the strongest bottlenecks have lower nucleotide diversity and higher individual inbreeding. In combination, our empirical analyses and individual-based simulations strongly suggest that the reintroductions led to the simultaneous accumulation of mildly deleterious mutations and purging of highly deleterious mutations.

## Results

### Genomic variation and inbreeding in ibex.
We analyzed 60 genomes covering seven species including the Alpine ibex (*C. ibex*), five additional wild goats and the domestic goat. Siberian ibex have the largest population size (~200,000), followed by Alpine and Iberian ibex (both ~50,000), Bezoar (~25,000), Markhor (~6000) and Nubian ibex (~2500)[39]. We found that Alpine ibex together with Iberian ibex and Markhor (*C. falconeri*) have exceptionally low genome-wide variation compared to closely related species (Fig. 1, Supplementary Fig. 1). All three species either underwent severe bottlenecks in the past century or are currently threatened (Fig. 1c, Supplementary Fig. 2, Supplementary Tables 1 and 2). In contrast to Alpine ibex suffering a bottleneck of ~100 individuals, Iberian ibex are thought to have been reduced to a census of ~1000 individuals historically[40] (Supplementary Table 1). In contrast, genome-wide diversity was highest in Siberian ibex (*C. sibirica*), which have large and relatively well-connected populations[41]. The genomes of some Alpine ibex, the Markhor and some domestic goat individuals contained more than 20% genome-wide runs of homozygosity (ROH) (Fig. 1d, Supplementary Figs. 3–5, Supplementary Table 3)[42]. Overall, there was clear genomic evidence that the near extinction and recovery of the Alpine ibex resulted in substantial genetic drift and inbreeding.

### Accumulation of deleterious mutations.
We analyzed all *Capra* genomes for evidence of segregating deleterious mutations (Supplementary Fig. 6). We restricted our analyses to coding sequences with evidence for transcriptional activity in Alpine ibex organs and high evolutionary conservation (i.e. GERP, see methods)[43] yielding a total of 370,853 SNPs (Supplementary Table 4). We found that across all seven *Capra* species 0.17% of these SNPs carried a highly deleterious variant with the majority of the highly deleterious variants incurring a stop-gain mutation, a further 19.1% carried a moderate impact variant and 33.1% carried a low impact variant (Supplementary Table 4). We used seven additional mutation impact scoring approaches (see Methods). The proportion of highly deleterious variants segregating within species was inversely correlated with nucleotide diversity (Fig. 1c and e, Supplementary Fig. 7a–d; Pearson 's product-moment correlation, $df = 5$, $r = -0.86$, $p = 0.012$). Hence, species with the smallest populations or the most severe population size reductions (i.e. Alpine ibex, Iberian ibex, and Markhor) showed an accumulation of deleterious mutations relative to closely related species.

### Deleterious mutations in Alpine and Iberian ibex.
Both Alpine and Iberian ibex experienced severe bottlenecks due to overhunting and habitat fragmentation. We first analyzed evidence for purifying selection using allele frequency spectra. We focused only on derived sites that were polymorphic in at least one of the two sister species (Fig. 2a, Supplementary Fig. 8). We found that frequency distributions of high and moderate impact mutations in Alpine ibex were downwards shifted compared to modifier (i.e. neutral) mutations, which strongly suggests purifying selection against highly deleterious mutations (Fig. 2b–d). Short indels (≤10 bp) in coding sequences revealed a similar downward shift (Supplementary Fig. 9). We found no comparable frequency shifts in Iberian ibex (Fig. 2b). This is consistent with purifying selection acting more efficiently against highly deleterious mutations in Alpine ibex compared to Iberian ibex. We repeated the analyses of the site frequency spectra (SFS) also for three alternative scoring systems (GERP, phyloP and phastCons) reporting phylogenetic conservation. Alpine ibex showed an excess of low frequency variants in the most deleterious mutation category for all three scores (Supplementary Fig. 10).

To test whether Alpine ibex showed evidence for purging of deleterious mutations, we calculated the relative number of

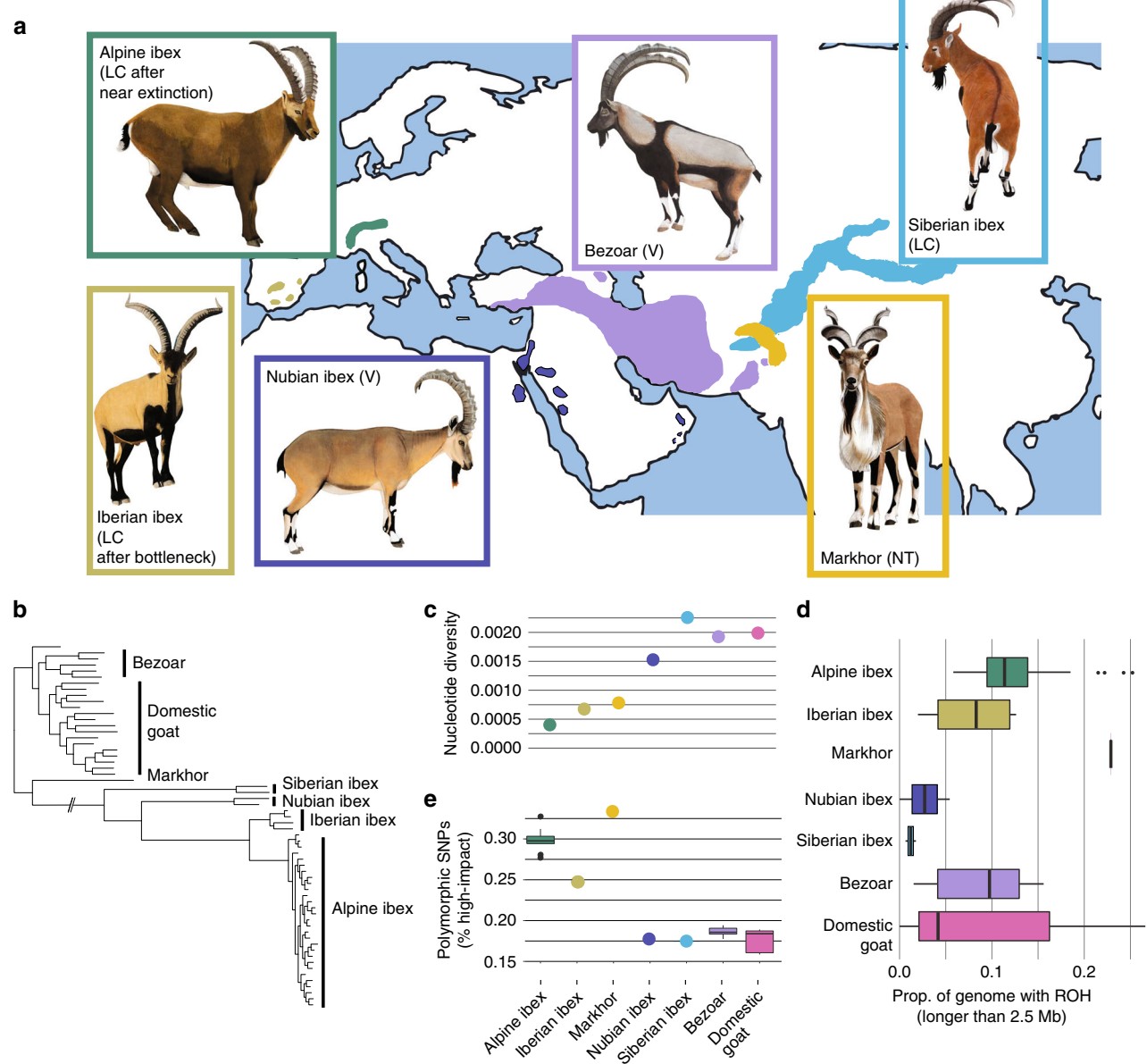

**Fig. 1 Genetic diversity and deleterious mutation load of ibex species. a** Geographical distribution and IUCN conservation status of ibex and wild goat species (LC: Least concern, V: Vulnerable, NT: Near threatened[39]). Sample sizes: Alpine ibex (*C. ibex*): N = 29, Iberian ibex (*C. pyrenaica*): N = 4, Bezoar (*C. aegagrus*): N = 6, Siberian ibex (*C. sibirica*): N = 2, Markhor (*C. falconeri*): N = 1, Nubian ibex (*C. nubiana*): N = 2. **b** Maximum likelihood phylogenetic analyses, (**c**) nucleotide diversity, (**d**) proportion of the genome with runs of homozygosity (ROH) longer than 2.5 Mb and (**e**) percentage of polymorphic sites within species that segregate highly deleterious mutations. Confidence intervals are based on downsampling to four individuals matching the sample size of Iberian ibex (100 replicates). Boxplots show the median, the 25th and 75th percentiles, Tukey whiskers (median ± 1.5 times interquartile range), and outliers (•). Source data are provided as a Source Data file.

derived alleles *Rxy*[28] for the different categories of mutations (Fig. 2d). We used a random set of intergenic SNPs for standardization, which makes *Rxy* robust against sampling effects and population substructure[28]. Low and moderate impact mutations (i.e. mildly deleterious mutations) showed a minor excess in Alpine ibex compared to Iberian ibex, indicating a higher load in Alpine ibex. In contrast, we found that highly deleterious mutations were strongly reduced in Alpine ibex compared to Iberian ibex (Tukey test, *p* < 0.0001, Fig. 2e). Strikingly, the proportion of SNPs across the genome segregating a highly deleterious mutation is higher in Alpine ibex (Fig. 1e), but *Rxy* shows that highly deleterious mutations have a pronounced downwards allele frequency shift in Alpine ibex

compared to Iberian ibex. Furthermore, the number of homozygous sites with highly deleterious mutations per individual was significantly lower in Alpine ibex than Iberian ibex (*t* test, *p* = 0.015, Fig. 2e). Individual allele counts at highly deleterious sites were also significantly lower in Alpine ibex compared to Iberian ibex (*t* test, *p* = 0.003). We assessed the robustness of the *Rxy* analyses using four additional mutation scoring methods (i.e. SIFT, REVEL, CADD, and VEST3). We found that the highest impact category had a consistent deficit in Alpine ibex compared to Iberian ibex (Supplementary Fig. 12). Together, this shows that highly deleterious mutations were substantially purged in Alpine ibex. We also found evidence for the accumulation of mildly deleterious mutations through genetic drift in Alpine ibex.

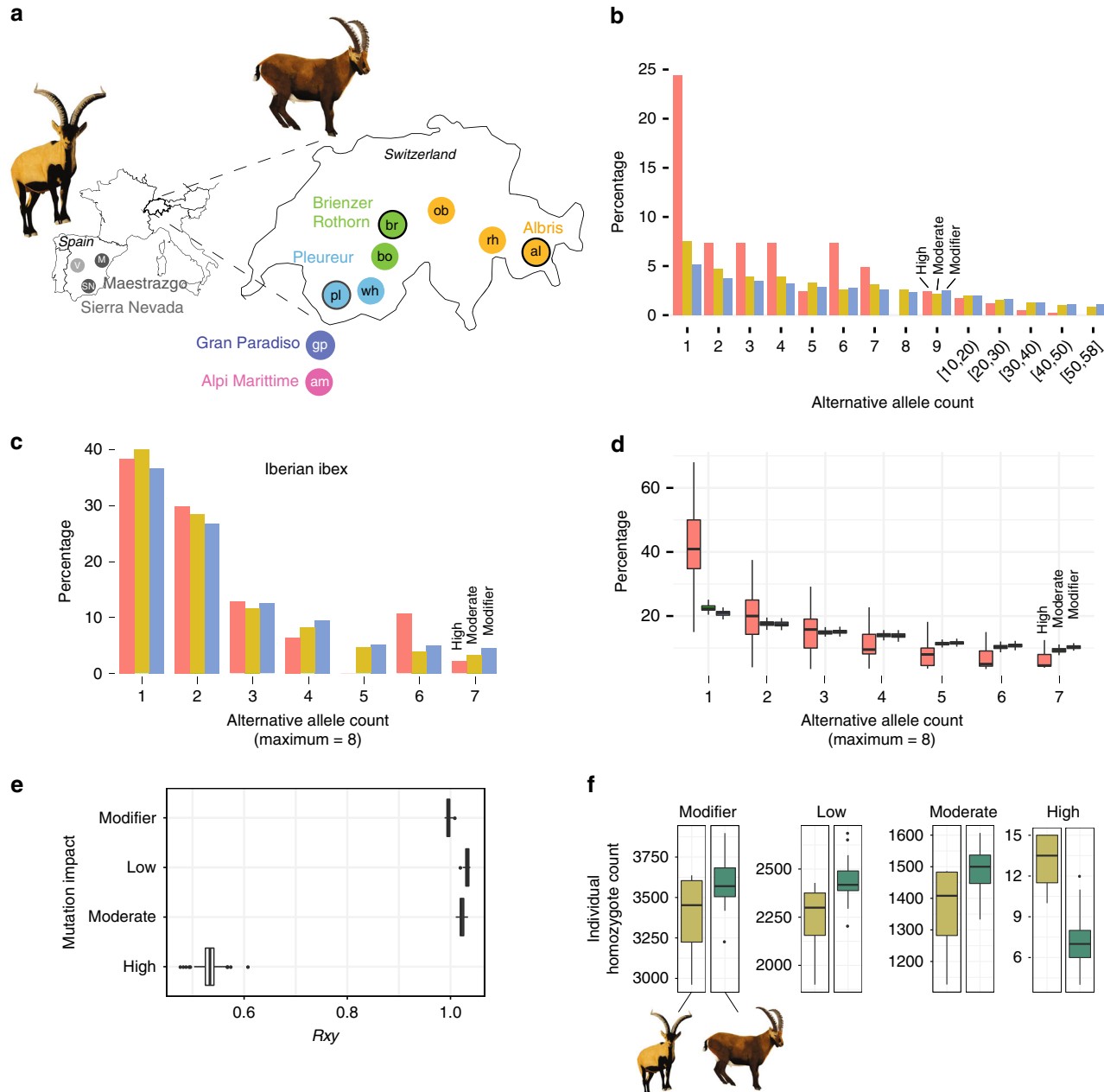

**Fig. 2 Deleterious mutations segregating in Alpine and Iberian ibex. a** Population sampling locations of Iberian ibex (left, gray circles) and Alpine ibex (right, colored circles). Each filled circle represents a population. Circles with a black outline indicate the first three reintroduced populations in Switzerland that were used for all subsequent population reintroductions of Alpine ibex. Colors associate founder and descendant populations (see also Fig. 3a). **b** Site frequency spectrum for neutral (modifier), mildly (moderate impact) and highly deleterious (high impact) mutations for Alpine ibex. Alternative allele counts > 10 are shown as mean counts per interval. **c** Site frequency spectrum for Iberian ibex. **d** Site frequency spectrum for Alpine ibex downsampled to four individuals matching the sample size of Iberian ibex (100 replicates). See Figure S11 for additional downsampling and jack-knifing analyses of data presented in (**b**–**d**). **e** $Rxy$ analysis contrasting Iberian with Alpine ibex across the spectrum of impact categories. $Rxy < 1$ indicates a relative frequency deficit of the corresponding category in Alpine ibex compared to Iberian ibex. $Rxy$ distributions are based on jack-knifing across chromosomes. All pairwise comparisons were significant except low vs. moderate (Tukey test, $p < 0.0001$). **f** Individual homozygote counts per impact category for Iberian (light green) and Alpine ibex (dark green). Boxplots elements are defined as in Fig. 1. Source data are provided as a Source Data file.

**Accumulation and purging of deleterious mutations.** Consistent with the fact that all extant Alpine ibex originate from the Gran Paradiso, this population occupies the center of a principal component analysis (Fig. 3a, b, Supplementary Fig. 13a, b;[38]). The first populations re-established in the Alps were clearly differentiated from the Gran Paradiso source population and showed reduced nucleotide diversity due to reintroduction bottlenecks[37] (Fig. 3a–c). These initial three reintroduced populations were

used to establish additional populations raising the total number of experienced bottlenecks to 3 or 4. The additional bottlenecks led to further loss of nucleotide diversity and genetic drift, as indicated by the increasing spread in the principal component analysis (Fig. 3a–c). An exceptional case constitutes the Alpi Marittime population, which was established through the translocation of 25 Gran Paradiso individuals of which only six successfully reproduced[44]. As expected from such an extreme

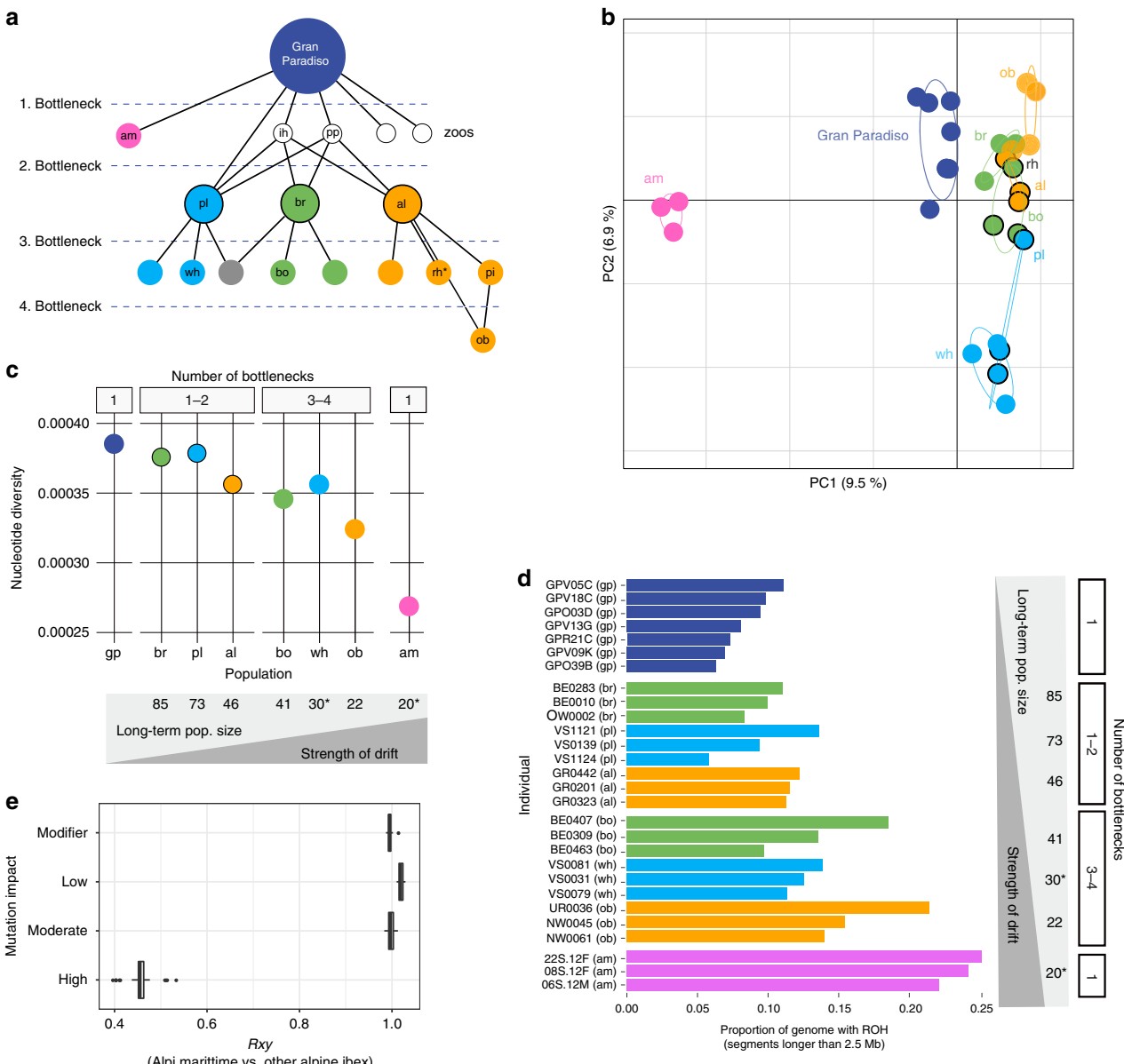

**Fig. 3 Genomic consequences of the Alpine ibex recolonization. a** The recolonization history and population pedigree of Alpine ibex. Locations include also zoos and the population Pilatus (pi), which was not sampled for this study. am: Alpi Maritime, gp: Gran Paradiso; ih: Zoo Interlaken Harder; al: Albris; bo: Bire Öschinen; br: Brienzer Rothorn; ob: Oberbauenstock; pl: Pleureur; rh: Rheinwald; wh: Weisshorn; pi: Pilatus; pp: Wildpark Peter and Paul. The gray circle represents a population that was founded from more than one population. Figure elements were modified from Biebach and Keller (2009) with permission. **b** Principal component analysis of all Alpine ibex individuals included in the study. **c** Nucleotide diversity per population. **d** Proportion of the genome within runs of homozygosity (ROH) longer than 2.5 Mb. **e** Rxy analysis contrasting the strongly bottlenecked Alpi Maritime population with all other Alpine ibex populations across the spectrum of impact categories. Rxy < 1 indicates a relative frequency deficit of the corresponding category in the Alpi Maritime population. Rxy distributions are based on jack-knifing across chromosomes. Circles with a black outline indicate the first three reintroduced populations in Switzerland that were used for all subsequent population reintroductions of Alpine ibex. Colors associate founder and descendant populations. Box plot elements are defined as in Fig. 1. Source data are provided as a Source Data file.

bottleneck, Alpi Maritime showed strong genetic differentiation from all other Alpine ibex populations and highly reduced nucleotide diversity (Fig. 3b, c;[45]). To compare the strength of drift experienced by different populations, we estimated long-term effective population sizes. For this, we used detailed demographic records spanning the near century since the populations were established[46,47]. We found that nucleotide diversity decreased with smaller long-term population size (Spearman's rank correlation, rho = 0.93, p = 0.007, Fig. 3c). We found the same trend for the individual number of heterozygous sites per kilobase (Supplementary Fig. 14). Populations with the lowest

harmonic mean population sizes also showed the highest levels of inbreeding. Genomes from the Gran Paradiso source population generally showed the lowest proportions of the genome affected by ROH, while reintroduced populations of lowest effective population size had the highest proportions of the genome affected by ROH (Pearson's product-moment correlation, df = 19, r = −0.70, p = 0.0004, Fig. 3d and Supplementary Fig. 3). The Rxy statistics showed a strong deficit of highly deleterious mutations in the Alpi Maritime population suggesting that the most deleterious mutations were efficiently purged (Fig. 3e and see below).

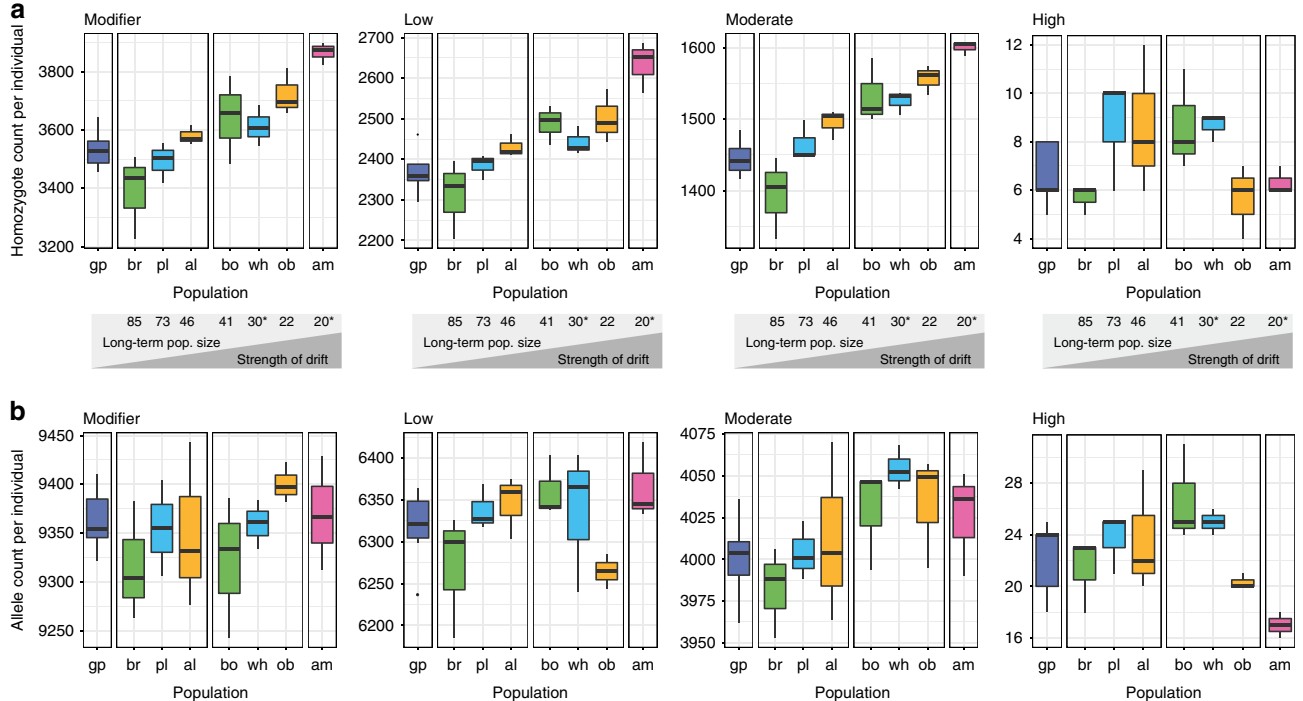

**Fig. 4 Impact of the recolonization on the individual mutation load. a** Homozygote counts and (**b**) allele counts per individual for each Alpine ibex population. The schematic between A and B indicates the harmonic mean of the census size of each population, which is inversely correlated with the strength of drift. * Estimated numbers. Colors associate founder and descendant populations (see also Fig. 3a). Box plot elements are defined as in Fig. 1. Source data are provided as a Source Data file.

Bottlenecks affect the landscape of deleterious mutations by randomly increasing or decreasing allele frequencies at individual loci. We find that individuals from populations that underwent stronger bottlenecks carry significantly more homozygotes for nearly neutral and mildly deleterious mutations (i.e. modifier, low and moderate impact mutations; Fig. 4a). In contrast, individuals showed no meaningful difference in the number of homozygotes for highly deleterious (i.e. high impact) mutations across populations. The stability in the number of homozygotes for highly deleterious mutations through successive bottlenecks despite a step-wise increase in the number of homozygotes for weaker impact mutations, supports that purging occurred over the course of the Alpine ibex reintroductions. We repeated the analyses using an alternative scoring of mutations based on the phylogenetic conservation of the region in which the mutations were found (i.e. Genomic Evolutionary Rate Profiling; GERP). The number of homozygotes for mutations in highly conserved regions showed a slight upwards trend still indicating purifying selection but not necessarily purging for this category of mutations (Supplementary Fig. 15). Interestingly, GERP scores (as well as other conservation-based scores) seem to perform well at distinguishing neutral mutations from mutations under selection, but less so to discriminate weakly from highly deleterious mutations. Hence, this suggests that the mutational impact (e.g. premature stop codons) rather than degree of conservation predicts whether purging is likely to occur. This suggests also that mean fitness should be more directly assessed using e.g. simulation datasets (see below). Because the above findings are contingent on a model where deleterious mutations are recessive, we also analyzed the total number of derived alleles per individual. We find a consistent but less pronounced increase in total number of derived alleles per individual for nearly neutral and mildly deleterious mutations (Fig. 4b). In contrast, the total number of derived alleles for highly deleterious mutations did not

correlate with the strength of bottleneck and was lowest in the most severely bottlenecked Alpi Marittime population (Fig. 4b), suggesting that the most deleterious mutations were purged in this population. The *Rxy* statistics showed a corresponding strong deficit in the Alpi Marittime population (Fig. 3e). We repeated the *Rxy* analyses using four additional mutation scoring methods (i.e. SIFT, REVEL, CADD, and VEST3) and found a consistent deficit of the highest impact category in the Alpi Marittime (Supplementary Fig. 12). Overall, we find evidence for more purging in the most bottlenecked Alpine ibex population.

We analyzed the impact of highly deleterious mutations by predicting the protein truncation using homology-based inferences. Focusing on mutations segregating in Alpine ibex, we found that nearly all mutations disrupted conserved protein family (PFAM) domains encoded by the affected genes (Fig. 5). This shows that highly deleterious mutations not only are altering the length of open reading frames but that evolutionarily conserved protein domains are affected by the mutations.

**Individual-based simulations under a realistic demographic scenario.** We further analyzed the impact of bottlenecks on different mutation classes using an individual-based forward simulation model parametrized with the demographic record[48] (Fig. 6a). The model realistically reproduced the reintroduction history and populations were parametrized with the actual founder size (Fig. 6a, Supplementary Fig. 16, Supplementary Table 1). We used *Rxy* to analyze the evolution of deleterious mutation frequencies through the reintroduction bottlenecks. The simulations showed a deficit of highly deleterious mutations, but an increase of mildly deleterious mutations after the reintroduction bottlenecks (Fig. 6b). This is consistent with our empirical evidence for purging during the species bottleneck (Fig. 2e). We computed genetic load defined as the mean individual fitness in

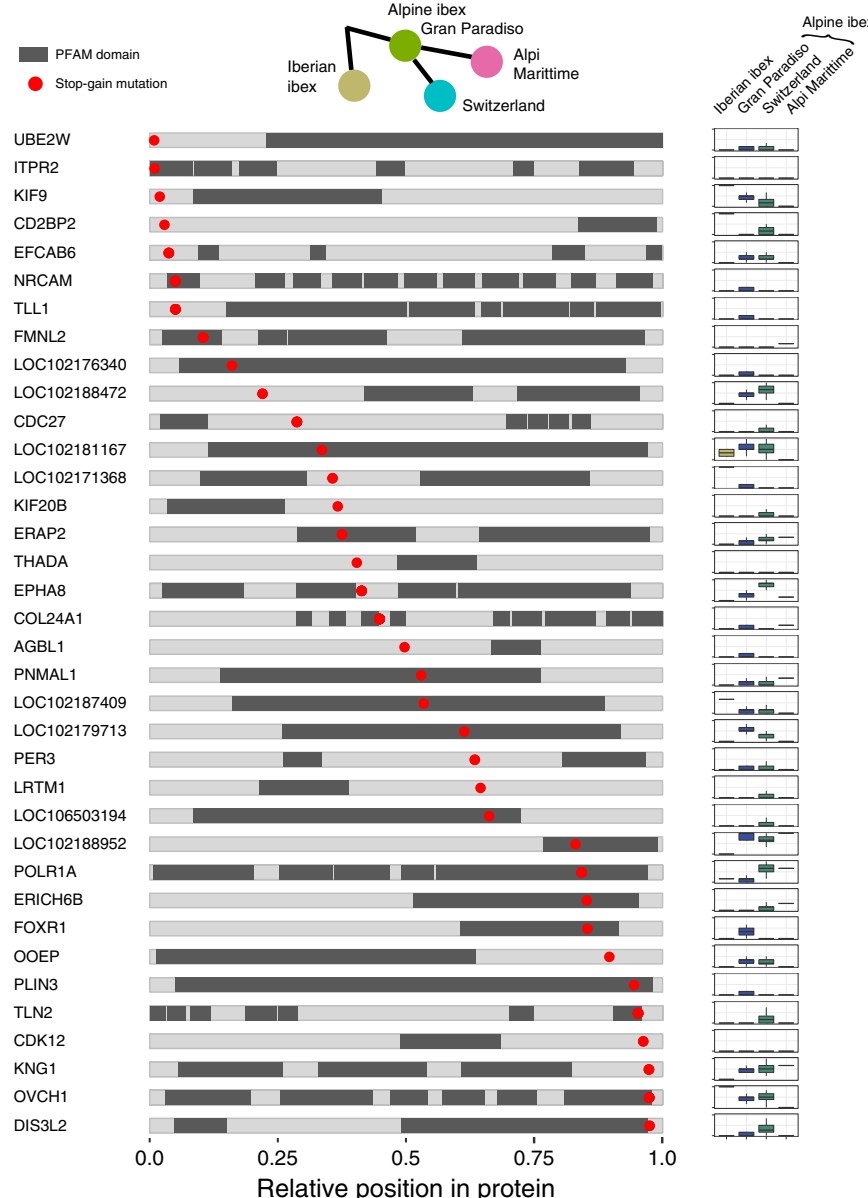

**Fig. 5 Homology-based inference of the impact of highly deleterious mutations.** The localization of protein family (PFAM) domains are highlighted in dark. Red dots indicate the relative position of a highly deleterious mutation segregating in Alpine ibex. The frequencies of highly deleterious mutations are summarized for Iberian ibex and three groups of Alpine ibex. Allele frequencies are shown as boxplots per group representing the outcome of downsampling to the smallest group (Alpi Marittime, $n = 3$) based on 100 replicates. The frequencies of highly deleterious mutations tend to be higher in Alpine ibex because mutation ascertainment was performed in this species. Box plot elements are defined as in Fig. 1. Source data are provided as a Source Data file.

females and found an increase in Alpine ibex following the species bottleneck (Supplementary Fig. 17). Hence, the accumulation of mildly deleterious mutations was reducing overall fitness despite purging. The simulations also supported purging at the level of individual populations as found in the extremely bottlenecked Alpi Marittime population (Fig. 3e). The number of derived mildly deleterious homozygotes increased with the strength of drift experienced by individual populations (Fig. 6c, Supplementary Figs. 18–21). In contrast, the median number of homozygote counts for high impact mutations were lower for Alpi Marittime than Gran Paradiso but not statistically significant (Fig. 6c, Mann–Whitney U, $p = 0.39$). The highly deleterious allele counts were significantly lower for Alpi Marittime compared to Gran Paradiso (Supplementary Fig. 19, Mann-Whitney U, $p = 0.001$). We also analyzed $Rxy$ for the simulation data and

found that high impact mutations were indeed relatively less frequent in Alpi Marittime compared to all other Alpine ibex populations (Fig. 6d) and compared to Gran Paradiso (Supplementary Fig. 22). Overall, the realistically parametrized model recapitulated all major empirical findings of mutation accumulation and purging across bottlenecks.

## Discussion
Ibex species with recently reduced population sizes accumulated deleterious mutations compared to closely related species. This accumulation was particularly pronounced in the Iberian ibex that experienced a severe bottleneck and Alpine ibex that went nearly extinct. We show that even though Alpine ibex carry an overall higher mutation burden than related species, the strong bottlenecks imposed by the reintroduction events purged highly

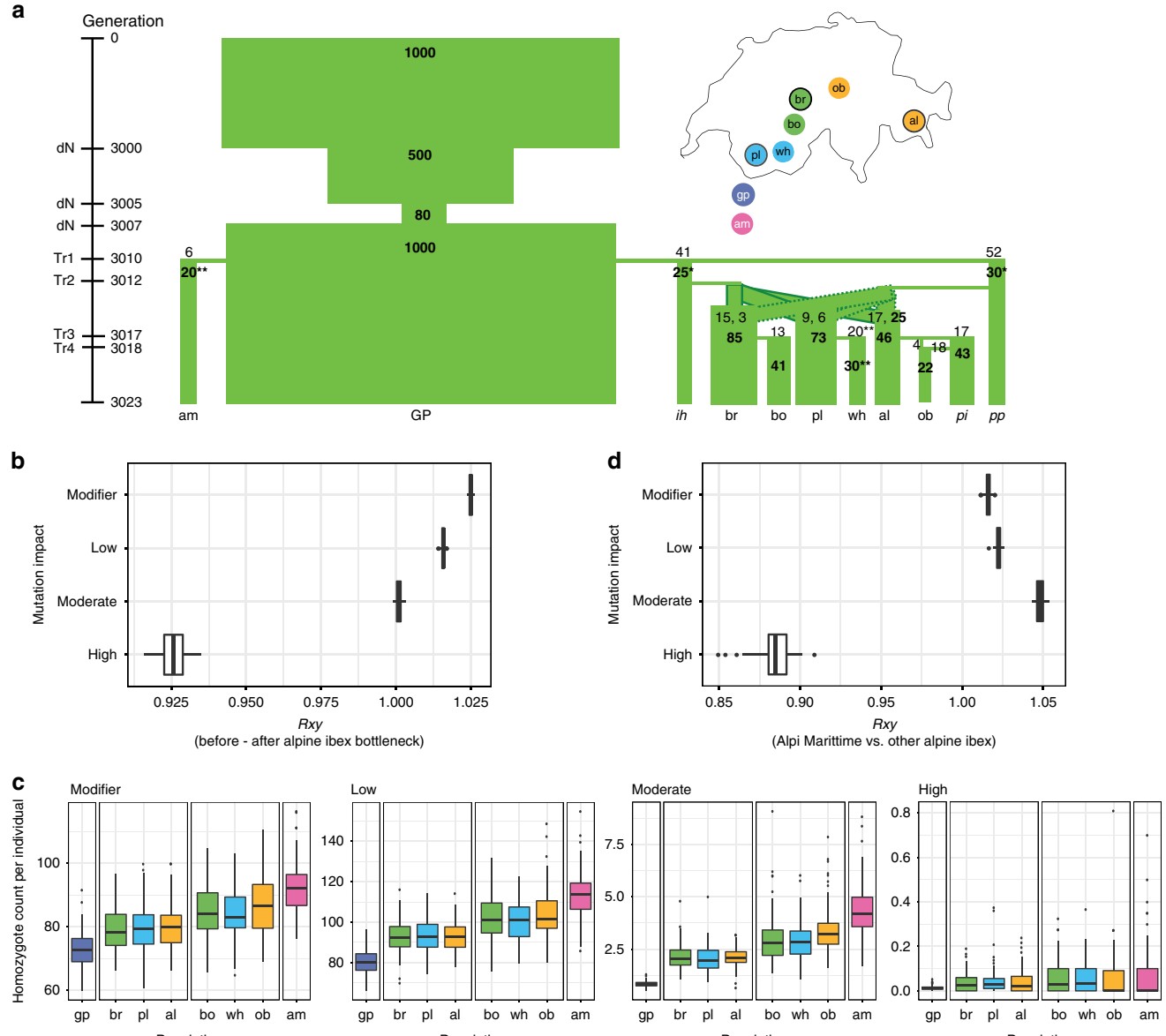

**Fig. 6 Individual-based simulations of the reintroduction history of Alpine ibex. a** The demographic model was parametrized using census data and historical records. Bold numbers show effective population sizes. Numbers not in bold indicate the number of individuals released to found each population. If a population was established from two source populations, the individual numbers are separated by commas. * Upwards adjusted harmonic means of the census size (historical records were ih = 16, Zoo Interlaken Harder, and pp = 20, Wildpark Peter and Paul). The adjustment was necessary to prevent extinction of zoo populations. ** Census numbers were estimated based on historical records of the population but no long-term census data were available. **b** Relative frequency comparison (*Rxy*) of Alpine ibex just before and after the species bottleneck and recolonization. **c** Individual homozygote counts per impact category. Boxplots summarize 100 population means across simulation replicates. Colors associate founder and descendant populations (see also Fig. 3a). **d** *Rxy* analysis contrasting the strongly bottlenecked Alpi Marittime population with all other Alpine ibex populations across the spectrum of impact categories. Box plot elements are defined as in Fig. 1. Source data are provided as a Source Data file.

deleterious mutations. Importantly, purging was only effective against highly deleterious mutations. Mildly deleterious mutations actually accumulated through the reintroductions. Hence, the overall number of deleterious mutations increased with bottleneck strength. This is consistent with the finding that population-level inbreeding, which is a strong indicator of past bottlenecks, is correlated with lower population growth rates in Alpine ibex[49].

Empirical evidence for purging in the wild is scarce[7,20,35]. Here, we show that a few dozen generations were sufficient to reduce the burden of highly deleterious mutations. Purging may occur widely in populations undergoing severe bottlenecks contingent

on populations surviving the consequences of inbreeding depression. Failure to purge under extreme bottlenecks can have severely deleterious consequences for wild populations such as shown for the Isle Royal wolves[50]. As predicted from theory, Alpine ibex reintroduction bottlenecks allowed mildly deleterious mutations to accumulate in contrast to highly deleterious mutations. Our empirical results are also in line with predictions that populations with an effective size below 100 individuals can accumulate a substantial burden of mildly deleterious mutations[5]. Such mutation load constitutes long-term extinction risks in contrast to short-term risks associated with highly deleterious mutations. The burden of deleterious mutations evident in

Iberian ibex supports the notion that even population sizes of ~1000 still accumulate mildly deleterious mutations. High loads of deleterious mutations have been shown to increase the extinction risk of a species[5]. Thus, conservation efforts aimed at keeping effective population sizes above a minimum of 1000 individuals[2] are critical for the long-term survival of managed species.

## Methods

**Genomic data acquisition**. DNA samples from 29 Alpine ibex, 4 Iberian ibex, 2 Nubian ibex, 2 Siberian ibex and 1 Markhor individuals were sequenced on an Illumina Hiseq2500 or Hiseq4000 to a depth of 15–38 (median of 17). Supplementary Table 2 specifies individual sampling locations. Libraries were produced using the TruSeq DNA Nano kit. Illumina sequencing data of 6 Bezoar and 16 domestic goat (coverage 6x–14x, median 12x) were generated by the NextGen Consortium (https://nextgen.epfl.ch). The corresponding raw data were downloaded from the EBI Short Read Archive: ftp://ftp.sra.ebi.ac.uk/vol1/fastq/.

**Read alignment and variant calling**. Trimmomatic v.0.36[51] was used for quality and adapter trimming before reads were mapped to the domestic goat reference genome (version CHIR1[52],) using Bowtie2 v.2.2.5[53]. MarkDuplicates from Picard (http://broadinstitute.github.io/picard, v.1.130) was used to mark duplicates. Genotype calling was performed using HaplotypeCaller and GenotypeGVCF (GATK, v.3.6[54,55]). VariantFiltration of GATK was used to remove single nucleotide polymorphisms (SNP) if: QD < 2.0, FS > 40.0, SOR > 5.0, MQ < 20.0, −3.0 > MQRandkSum >3.0, −3.0 > ReadPosRankSum >3.0 and AN < 62 (80% of all Alpine ibex individuals). Indels up to 10 bp were also retained and filtered using the same filters and filter parameters, except for not including the filter MQRankSum, because this measure is more likely to be biased for indels of several base pairs. Filtering parameters were chosen based on genome-wide quality statistics distributions (see Supplementary Figs. 23–40). Variant positions were independently validated by using the SNP caller Freebayes (v1.0.2-33-gdbb6160[56]) with the following settings:–no-complex –use-best-n-alleles 6 –min-base-quality 3 –min-mapping-quality 20 –no-population-priors –hwe-priors-off.

To ensure high-quality SNPs, we only retained SNPs that were called and passed filtering using GATK, and that were confirmed by Freebayes. Overall, 97.5% of all high-quality GATK SNP calls were confirmed by Freebayes. This percentage was slightly lower for chromosome X (96.7%) and unplaced scaffolds (95.2%). We tested whether the independent SNP calls of GATK and Freebayes were concordant and we could validate 99.6% of the biallelic SNPs. We retained genotypes called by GATK and kept SNPs with a minimum genotyping rate of 90% for all further analysis. The total number of SNPs detected was 59.5 million among all species. Per species, the number of SNPs ranged from 21.9 million in the domestic goat ($N = 16$) to 2.0 million in Markhor ($N = 1$, Supplementary Table 2).

**RNA-seq data generation**. Tissue samples of a freshly harvested Alpine ibex female were immediately conserved in RNA*later* (QIAGEN) in the field and stored at −80 °C until extraction. The following ten organs were sampled: retina/uvea, skin, heart, lung, lymph, bladder, ovary, kidney, liver, and spleen. RNA was extracted using the AllPrep DNA/RNA Mini Kit from Qiagen following the manufacturer's protocol. Homogenization of the samples was performed using a Retsch bead beater (Retsch GmbH) in RLT plus buffer (Qiagen). RNA was enriched using a PolyA enrichment protocol implemented in the TruSeq RNA library preparation kit. Illumina sequencing libraries were produced using the Truseq RNA stranded kit. Sequencing was performed on two lanes of an Illumina Hiseq4000.

**Genetic diversity and runs of homozygosity**. Genetic diversity measured as individual proportion of heterozygous sites and nucleotide diversity were computed using vcftools[57]. Runs of homozygosity were called using BCFtools/RoH[58], an extension of the software package BCFtools, v.1.3.1. BCFtools/RoH uses a hidden Markov model to detect segments of autozygosity from next generation sequencing data. Due to the lack of a detailed linkage map, we used physical distance as a proxy for recombination rates with the option −M and assuming 1.2 cM/Mb following sheep recombination rates[59]. Smaller values for −M led to slightly longer ROH (Supplementary Figs. 3–5). Because of small per population sample size, we decided to fix the alternative allele frequency (option -AF-dflt) to 0.4. Estimates for the population with the largest sample size (Gran Paradiso, $N = 7$) were very similar if actual population frequencies (option –AF-estimate sp) were used (Supplementary Figs. 4 and 5). Option –viterbi-training was used to estimate transition probabilities before running the HMM. Running the analysis without the option –viterbi-training led to less but longer ROH (Supplementary Figs. 3–5). ROH were also estimated using PLINK (v1.90b5, https://www.cog-genomics.org/plink/) with the following settings:–homozyg-window-het 2, –homozyg-window-missing 5, –homozyg-snp 100, –homozyg-kb 500, –homozyg-density 10, –homozyg-gap 100, –homozyg-window-threshold .0. ROH estimates based on PLINK were overall

slightly lower but the qualitative trends hold among species and population (Supplementary Figs. 41 and 42).

**Identification of high-confidence deleterious mutations**. Three lines of evidence were used to identify high-confidence deleterious mutations. First, variants leading to a functional change are candidates for deleterious mutations. We used snpEff[60] v.4.3 for the functional annotation of each variant. The annotation file ref_CHIR_1.0_top_level.gff3 was downloaded from: ftp://ftp.ncbi.nlm.nih.gov/genomes/Capra_hircus/GFF and then converted to gtf using gffread. Option −V was used to discard any mRNAs with CDS having in-frame stop codons. SnpEff predicts the effects of genetic variants (e.g. stop-gain variants) and assesses the expected impact. The following categories were retrieved: high (e.g. stop-gain or frameshift variant), moderate (e.g missense variant, in-frame deletion), low (e.g. synonymous variant) and modifier (e.g. exon variant, downstream gene variant). In the case of overlapping transcripts for the same variant, we used the primary transcript for further analysis. A total of 49.0 % of all detected SNPs were located in intergenic regions, 43.2 % in introns, 6.5% down- and upstream of genes. A total of 0.7% of variants were within CDS, of which ~60% were synonymous and ~40% were missense variants. Overall, 0.002% were stop-gain mutations.

Protein sequences were annotated using InterProScan v.5.33 by identifying conserved protein family (PFAM) domains[61].

Second, we assessed the severity of a variant by its phylogenetic conservation score. A non-synonymous variant is more likely to be deleterious if it occurs in a conserved region of the genome. We used GERP conservation scores, which are calculated as the number of substitutions observed minus the number of substitutions expected from the species tree under a neutral model. We downloaded GERP scores (accessed from http://mendel.stanford.edu/SidowLab/downloads/gerp), which have been computed for the human reference genome version hg19. The alignment was based on 35 mammal species but did not include the domestic goat (see https://genome.ucsc.edu/cgi-bin/hgTrackUi?db=hg19&g=allHg19RS_BW for more information). Exclusion of the focal species domestic goat is recommended for the computation of conservation scores, as the inclusion of the reference genome may lead to biases[62]. In order to remap the GERP scores associated to hg19 positions to the domestic goat reference genome positions, we used liftOver (hgdownload.cse.ucsc.edu, v.287) and the chain file downloaded from hgdownload-test.cse.ucsc.edu/goldenPath/capHir1.

Third, we ascertained support for gene models annotated in the domestic goat genome with expression analyses of Alpine ibex tissue samples. We included expression data from 10 organs of an Alpine ibex female (see RNA-seq data section above) to assess expression levels of each gene model. Quality filtering of the raw data was performed using Trimmomatic[51] v.0.36. Hisat2[63] v.2.0.5 was used to map the reads of each organ to the domestic goat reference genome. The mapping was run with option –rna-strandness RF (stranded library) and supported by including a file with known splice sites (option –known-splicesite-infile). The input file was produced using the script hisat2_extract_splice_sites.py (part of hisat2 package) from the same gtf file as the one used for the snpEff analyis (see above). For each organ, featureCounts[64] (subread-1.5.1) was used to count reads per each exon using the following options: -s 2 (reverse stranded) –f (count reads at the exon level), –O (assign reads to all their overlapping features), –C (excluding read pairs mapping to different chromosomes or the same chromosome but on a different strand). The R package edgeR[65] was used to calculate FPKM (Fragments Per Kilobase Of Exon Per Million Fragments Mapped) per each gene and organ. For variant sites that were included in more than one exon, the highest FPKM value was used. We found that 16,013 out of 17,998 genes showed transcriptional activity of at least one exon (FPKM > 0.3). Overall 166,973 out of 178,504 exons showed evidence for transcription. In a total of 1928 genes, one or more exons showed no evidence for transcription. Retained SNPs were found among 118,756 exons and 17,685 genes. Overall 611,711 out of 677,578 SNPs were located in genes with evidence for transcription.

Deleterious mutations are assumed to be overwhelmingly derived mutations. We used all ibex species except Alpine and Iberian ibex as an outgroup to define the derived state. For each biallelic site, which was observed in alternative state in Alpine ibex or Iberian ibex, the alternative state was defined as derived if its frequency was zero in all other species (a total of 44,730 autosomal SNPs). For loci with more than two alleles, the derived state was defined as unknown. For comparisons among all species, we only used the following criteria to select SNPs (370'853 biallelic SNPs retained): transcriptional activity (FPKM > 0.3 in at least one organ) and GERP > −2. The minimum GERP score cutoff was set to retain only high-quality chromosomal regions following previously established practice e.g.[23]. We further followed the following categorizations adopted for human populations to identify moderate to highly deleterious mutations e.g.:[23] −2 to +2 is considered neutral or near neutral, 2–4 considered as moderate, 4–6 as large and >6 as extreme effects. We also required a minimal distance to the next SNP of 3 bp to avoid confounding effects of potential multi-nucleotide polymorphisms (MNPs).

**Population genetic analyses**. SFSs were calculated using the R packages *plyr* and *dplyr*. SFS analyses were performed for SnpEff and GERP categories and two additional conservation scores: phyloP[66] and phastCons[67]. We chose a cutoff of 1 to distinguish conserved from less conserved sites. In the case of phyloP, sites with

a score above 1 were defined as conserved. For phastCons, sites with a score equal to 1 (the maximum observed value) were considered as conserved.

For individual counts of derived alleles or homozygotes, we used all biallelic sites polymorphic either in Alpine or Iberian ibex (or both) for which the derived state was known with a maximal missing rate per locus of 10%. We retained all sites matching these criteria for any downstream analyses even if a particular site was not polymorphic in any given population. The effective rate of missing data per locus was between 0.03–0.07% (Supplementary Fig. 43) and no correlation was found between missing rate per population and counts. We found no qualitative differences if we included only loci with a 100% genotyping rate (Supplementary Fig. 44).

We calculated the relative number of derived alleles $Rxy$[28] for the different categories of mutations. $Rxy$ compares the number of derived alleles found at sites within a specific category. Following[9,28], we used a random set of 65592 intergenic SNPs for standardization, which makes $Rxy$ robust against sampling effects and population substructure. We performed the $Rxy$ analysis for the four SnpEff categories as well as four additional mutation scoring methods: SIFT[68], REVEL[69], CADD[70], and VEST3[71]. Human scores mapped to hg38 chromosomal positions were retrieved using the web interface of the Variant Effect Predictor (VEP) by ensemble[72]. The scores were mapped to chromosomal positions in the domestic goat reference genome using liftOver (http://hgdownload.cse.ucsc.edu, v.287) with the chain file accessed from http://hgdownload-test.cse.ucsc.edu/goldenPath/capHir1. As we applied these scores outside of humans, we did not have pathological evidence underpinning score cut-offs for deleteriousness. We conservatively used score cut-offs proposed as best-practices by the tool developers (Revel: 0.75, CADD: 20, SIFT: 0.05) or used a very conservative percentile (99%, 0.91 VEST3).

**Individual-based simulations with Nemo**. Individual-based forward simulations were run using the software Nemo[48] v.2.3.51. A customized version of aNEMOne[73] was used to prepare input files for parameter exploration. The sim.ini file for the final set of parameters run in 100 replicates is available as Supplementary Note 1. All populations relevant for the founding of the populations under study were included in the model. See Fig. 6a for the simulated demography, which was modeled with the actual founder numbers (assuming a sex-ratio of 1:1), while the translocations were simplified into four phases (data from ref. [48], DRYAD entry https://doi.org/10.5061/dryad.274b1 and[46]). The harmonic mean of the population census from the founding up to the final sampling year (2007) was used to define the population carrying capacity. Mating was assumed to be random and fecundity (mean number of offspring per female) set to five. The selection coefficients of 5000 biallelic loci subject to selection against deleterious mutations were drawn from a gamma distribution with a mean of 0.01 and a shape parameter of 0.3 resulting in $s$ <1% for 99.2% of all loci[74] (Supplementary Fig. 45). Based on empirical evidence, we assumed a negative relationship between $h$ and $s$[75]. We used the exponential equation $h = \exp(-51*s)/2$ with a mean $h$ set to 0.37 following[76]. We assumed hard selection acting at the offspring level. In addition to the 5000 loci under selection, we simulated 500 neutral loci. Recombination rates among each neutral or deleterious locus was set to 0.5. This corresponds to an unlinked state. Initial allele frequencies for the burn-in were set to $\mu / h * s = 0.0014$ (corresponding to the expected mean frequency at mutation-selection balance[77]). Mutation rate $\mu$ was set to 5e−05 and deleterious mutations were allowed to back-mutate at a rate of 5e−07.

A burn-in of 3000 generations was run with one population ($N = 1000$) representing the entire species. The burn-in was designed to be too short to reach mutation-drift-selection equilibrium to retain segregating deleterious mutations. The number of segregating deleterious mutations and heterozygosity at deleterious mutation loci changed rapidly early in the burn-in, reached a plateau at ~3000 generations before slowly decreasing. $N$ was reduced to $N = 500$ for five generations before a brief, two generation bottleneck of $N = 80$. At generation 3007, the population recovered to $N = 1000$ and three generations later the reintroduction was started with the founding of the two zoos Interlaken Harder (ih) and Peter and Paul (pp). The founding of new populations was modeled by migration of offspring into an empty patch.

The zoo ih (Interlaken Harder) and several populations did not survive all replicates of the simulations. Extinction rates were as follows: ih (Zoo Interlaken Harder) 84%, bo (Bire Öschinen) 3%, wh (Weisshorn) 3%, ob (Oberbauenstock) 9%, am (Alpi Marittime) 14%, and pil (Pilatus) 2%. The high extinction rate of the zoo Interlaken Harder did not affect the outcome of the simulations. The extinctions were a result of the strong reduction in population size during the founding and occurred always after the founding (see also Supplementary Fig. 16). The extinctions of the reintroduced populations did not affect the estimates of derived allele counts but reduced sample sizes and, hence, affected the variance of estimators. Genetic load information was retrieved using the *delet* statistics option and is defined in Nemo as the mean realized fitness ($L = 1 - W_{mean}/W_{max}$, where $W_{max}$ is the maximum number of surviving offspring per female computed from her deleterious mutations).

**Ethics statement**. This work complies with institutional guidelines and was authorized by the Swiss animal experimentation permit no. GR_6/2007.

**Reporting summary**. Further information on research design is available in the Nature Research Reporting Summary linked to this article.

## Data availability

Raw whole-genome sequencing data produced for this project was deposited at the NCBI Short Read Archive under the Accession nos. SAMN10736122-SAMN10736160 (BioProject PRJNA514886). The whole-genome data produced by the NexGen Consortium (*Capra hircus* accessions: ERR470105, ERR470101, ERR313212, ERR313211, ERR313204, ERR297229, ERR313206, ERR405774, ERR405778, ERR315778, ERR318768, ERR246140, ERR340429, ERR246152, ERR345976, *Capra aegagrus* accessions: ERR340334, ERR340340, ERR340333, ERR340331, ERR340335, ERR340348) was downloaded from [ftp://ftp.sra.ebi.ac.uk/vol1/fastq]. Raw RNA sequencing data produced for this project was deposited at the NCBI Short Read Archive under the Accession nos. SAMN10839218-SAMN10839227 (BioProject PRJNA517635). The source data underlying Figs. 1–6 are provided in a Source Data file.

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

## Acknowledgements

The authors thank the following organizations and colleagues who contributed samples to this project. Iris Biebach, the Swiss hunting authorities of the cantons of Bern, Nidwalden, Obwalden, Uri, Graubünden and Wallis; the Gran Paradiso National Park (Alice Brambilla) and the Alpi Maritime National Park (Laura Martinelli), Sebastien Regnaut and Richard Kock, Zoological Society of London, Christian Siegenthaler, Ruedi Kunz and Samer Angelone-Alasaad. We are thankful to Glauco Camenisch and Kasia Sluzek, who provided access to Alpine ibex RNAseq datasets. We thank Laurent Excoffier, Stephan Peischl, Kimberly Gilbert, Heidi Lischer, Stefan Wyder, Thomas Wicker, Alan Brelsford, Jessica Purcell, Sarah P. Otto, Andreas Wagner, Sam Yeaman and Nicolas Perrin for helpful advice and comments on previous versions of the manuscript. We are grateful for drawings by Nadine Coline of the Zoological Museum of Zürich. This work was supported by the University of Zurich through a University Research Priority Program "Evolution in Action" pilot project grant and the Swiss Federal Office for the Environment. DC and CG were supported by the Swiss National Science Foundation (grant 31003A_173265 and 31003A_182343, respectively). This study makes use of data generated by the NextGen Consortium, which was supported by grant agreement number 244356 of the European Union's Seventh Framework Programme (FP7/2010-2014).

## Author contributions

C.G., L.F.K., D.C. conceived and designed the study. C.G. acquired and analyzed the data. C.G., F.G., L.F.K., D.C. interpreted the data. C.G. and L.F.K. provided funding. C.G. and D.C. wrote the manuscript with input from the other authors.

## Competing interests

The authors declare no competing interests.
