## [Peer Review File · Nature Communications]

Reviewers' Comments:

Reviewer #1:

Remarks to the Author:

The manuscript by Grossen et al. examines patterns of deleterious mutations in six ibex species. They find several interesting patterns, including more putatively weakly deleterious and nearly neutral alleles and homozygous genotypes in more bottlenecked populations. However, strongly deleterious mutations appear to be reduced in the more bottlenecked populations, consistent with the effects of purging of deleterious mutations. This study combines whole genome sequence data with detailed population genetic modeling. As such, it takes a comprehensive approach to addressing important questions in an intriguing study system.

I have a number of major comments with some of the technical aspects of the manuscript:

1) The authors examine 6 populations. This is indeed a strength of the paper. However, the number of genomes sampled from each population varies considerably. For example, 29 Alpine ibex were sampled, while only 4 Iberian ibex were sampled. Some of the analyses presented here are sensitive to sample size and, as such, it is hard to draw meaningful conclusions with such disparate sample sizes. The authors should downsample the populations with a large number of individuals to make things more comparable across populations. Specifically, the results in Figure 1E, the SFSs in Figure 2, and the frequency plots in Figure 5 will be especially sensitive to sample size. It's not really meaningful to compare SFS of different sample sizes to each other.

2) SFS comparison in Figure 2: The authors conclude the fact that in the Alpine Ibex, more deleterious mutations are skewed more toward low frequency constitutes evidence of purging. I do not think that this difference in the SFS constitutes evidence of purging. Rather, this skew in the SFS would indicate that the high impact mutations are under more purifying selection than the moderate or modifier mutations. This finding is what one would expect to see for deleterious mutations. It does not imply purging. The fact that this pattern is not seen with the Iberian Ibex could be due to the differences in sample size (see my previous comment). Further, Figure 2 uses the proportional SFS here (ie the proportions of segregating SNPs at different frequency) rather than the absolute SFS (ie the numbers of SNPs at different frequencies). Thus, this figure does not show whether there are more or less high impact SNPs. It's just saying that of the SNPs that are there, they are skewed toward lower frequency. We would expect fewer strongly deleterious mutations to be segregating at all and this would not be picked up in the proportional SFS.

3) Comparisons of derived alleles and homozygous genotypes across populations can be sensitive to missing data. How was missing data dealt with here? Are levels of missing data similar across populations?

4) How were fixed sites dealt with in the counts of homozygous derived genotypes across populations? Specifically, if there was a site where all individuals in a population were homozygous for the derived allele, yet in other population, there were 2 alleles segregating, how would this site be counted? What if there are sites that are fixed for different alleles in different populations? How were these counted?

5) Lines 137-144 (and associated figures): The authors compare the numbers of homozygous deleterious mutations and derived alleles across different populations. Are these differences statistically significant? Some sort of statistical test should be presented here. Similarly, in the Ryx plots (e.g. Figure 2E), what are the distributions over? How was statistical significance assessed here?

6) Figure 6 and forward simulations: I like how the authors include forward simulations with realistic

demographic parameters. However, on line 251, the authors indicate that "Alpine ibex carry an overall higher mutation burden than related species". This can be directly computed from the model. Specifically, the mean fitness of each of the populations can be computed given the numbers of segregating deleterious mutations, their selection coefficients, and their dominance coefficients. It would be good to compute this actual genetic load from the simulations and see how it compares. There can be differences between the load computed from simulations vs. what's seen empirically when counting mutations (e.g. see Pedersen et al. 2017 Genetics).

7) Simulations: Do the simulated Alpi Maritime have fewer homozygotes for high impact mutations than do the gp? It's hard to tell from Figure 6c. This finding is important for demonstrating purging.

8) It would be good to better related Figure 6B-D to the empirical patterns. Right now it's hard to see which simulated pattern goes with which empirical pattern.

9) Figure S13: Here when GERP scores are used to predict which mutations are likely to be deleterious, in the 6+ category, it does not appear that "am" have fewer homozygous genotypes or derived alleles. Based on this figure there does not appear to be any evidence of purging (ie the more inbred or bottlenecked population having fewer deleterious alleles). Some reasons for this discrepancy with Figure 4 in the main text should be presented. One possibility is that GERP is not doing a good job at distinguishing strongly deleterious mutations from more moderately deleterious ones. I think this needs to be highlighted and discussed more in the main text. This also motivates the need to consider examining mean fitness directly from the simulations, in addition to relying on bioinformatics predictions of fitness effects on empirical data.

Additionally, I have several minor suggestions to improve the paper:

1) Lines 55-59: There is some conflating between the effects of genetic drift allowing weakly deleterious mutations to increase in frequency and the effects of inbreeding (ie mating between close relatives) throughout the manuscript. Though they can often occur together, especially in small populations, these are distinct processes, and can have different effects on weakly deleterious vs. strongly deleterious mutations as well as additive vs. recessive mutations. Thus, I would urge the authors to use the terminology more carefully and precisely throughout the paper. Also, line 162 is problematic for this same reason.

2) Line 65: Some additional references to cite here include: Do et al. 2015 Nature Genetics; Lohmueller 2014 Curr Opin Genet Dev; Fu et al 2014 AJHG.

3) Figure 5: I like this figure for showing the locations of the highly deleterious mutations in the proteins. However, I'm not sure what I'm supposed to conclude from the panel at the right showing the frequencies of these mutations in the different populations. More context and interpretation should be provided.

4) Lines 257-259: Here the authors seem to imply that bottlenecks may be good because they allow for purging of deleterious mutations. However, this is conditional on the population surviving and not going extinct due to inbreeding depression in the meantime. The Isle Royale wolves offer a counter example. Here, small population size combined with strong inbreeding led to severe inbreeding depression. Robinson et al 2019 (Science Advances) provides a description and analysis of this system.

5) Lines 309: The authors detect ROHs using BCFtools. As this method relies on allele frequencies, I would suggest that it would be good to also compare to the ROHs seen using other programs, like

PLINK. (I do agree that BCFtools is the preferred approach, but as a sanity check, it would be good to see whether the broad qualitative trends hold with PLINK too).

6) Line 396: Is this initial frequency of the deleterious mutations at mutation-selection balance used at the start of the burn in?

7) Figure S6: The GERP score thresholds presented here are a bit confusing. The figure says "only retain high quality deleterious mutations: $GERP > -2$ ". Many mutations with $GERP > -2$ won't be deleterious. Also, a few lines down in this figure, it would be better to order the GERP and the snpEFF functional predictions in the same order (e.g. low to high for both).

Reviewer #2:

Remarks to the Author:

The authors sequenced and analyzed 60 complete genomes of six ibex species and the domestic goat to investigate the relation between bottleneck strength and mutation load. These ibex species experienced the most dramatic bottlenecks among the successfully restored species. For instance, Alpine ibex were reduced to ~100 individuals in the 19th century and now about 50'000 individuals have been re-established in Alps. They concluded that historic bottlenecks, temporary reductions in population size, induced by human activity impacts levels of genome-wide variation and caused both relaxed selection and purging that reshaped landscape of deleterious mutation load. This study also proposed that for ensuring long-term survivability of endangered species, conservation efforts should model bottleneck intensity. In their study, defining the phenotypic effects of variants is an important issue, and the authors defined the mutational effects using snpEff toolbox and GERP conservation scores for genes with evidence for transcription. However, two major problems exist in defining high-confidence deleterious mutations.

Major comments:

1. Four categories proposed by snpEff are HIGH, MODERATE, LOW, and MODIFIER, which is a simple estimation of putative impact/deleteriousness for the Sequence Ontology terms. For instance, snpEff categories stop-gain, frameshift or start-lost variants as high putative impact, and missense variant, in-frame deletion and insertion with moderate impact. As we know, many missense mutations can cause diseases. Generally, the possibility or the fraction of variants with high putative impact to be deleterious mutations should be higher than moderate or low impact mutations. However, the sequencing results show that the number of high impact mutations by snpEff is much less than other categories. In Table S4, the number of SNPs is 614 and 70,873 classified as high and moderate respectively for all ibex species. Even though 1% of moderate variants suggested by snpEff have the possibility to be deleterious mutations, the number of it is about 708. Therefore, the authors should provide a more rigorous definition for the phenotypic effects of variants. For instance, using several other popular prediction methods (such as Revel, Vest or CADD) to find the deleterious mutations in moderate category and then do further analysis.

2. The authors assessed the severity of a variant using GERP conservation score. There are another two popular conservation scores, phyloP and phastCons, and I suggest the authors to validate their conclusion using one of these scores.

Minor comments:

1. I cannot find GERP scores from <http://mendel.stanford.edu/SidowLab> that the authors provided.
2. Please illustrate why the authors choose the cutoffs of $GERP > -2$ and the minimal distance to the next SNP of 3bp.

Reviewer #3:

Remarks to the Author:

Authors here report a genomic study on the Alpine Ibex in the Alps. This species can be seen as a model species for investigating the role of bottlenecks on demography as well as shaping genomic structure. Thus the study set provides a unique opportunity for investigating the role of purging under in natural populations when the species has been facing strong bottlenecks. So far only very few studies have actually been undertaken trying to identify the role of selection and estimating the mutation load in natural systems and the current study does fill an important gap. The strength of the manuscript is the combination of a well documented demographic history and thus the combination of population sizes, bottleneck history and the genomic analyses. Relating purging to the different history and population scenarios definitely is highly relevant for a conservation perspective. Analyses are novel and appropriate and the appendix provides a massive amount of background data. Thus I do think the paper will make a significant contribution not only to the field of Conservation Biology and Genetics but also to understand evolutionary processes in the wild.

I have two points, I think, authors should address, though:

Sample size and distribution is not equal for all investigated populations. While I understand that the populations in the Alps have been in the focus also of previous studies and thus most material comes from this population I wonder how representative the sampled genomes of the other species are. Some have only two genomes and probably are opportunistically be sampled, but as information is missing on population sizes of the other Ibex species (Table S1A and TableS1B are only for Alps and Iberian Ibex) it is a bit unclear to me, if those few samples from the other populations are good representatives and if and how authors have dealt with that fact. I would also recommend to add not only rough population sizes for the other species, but also to include sampling dates for all genomes in the appendix.

line32: sentence reads as if something is missing here

line46/47: at several places I found the use of tenses inconsistently, as e.g. in the first and second sentence of the introduction – please check.

line 55: “purging.Purging” more a stylistic remark, but could be reworded

Figure 1: line 78: in addition to the IUCN conservation status it would be good to add rough population sizes in the boxes for each species as well

line 80/87: how do sample sizes match – in the figure N=29 is mentioned for Alpine Ibex, in the text you refer to 60 genomes (which is probably rather a matter of wording, but I first was confusing that with 60 Alpine Ibex samples)

line116/117: So what is the explanation for this? I did not find that afterwards.

line120: I can understand the rationale of contrasting Iberian Ibex to Alpine Ibex in that graph, however, I would like to see a comparison to the other species (probably in appendix and domestic goat species as well)

line143/144: is this true also when you analyse the different alpine Ibex populations separately?

line259/261: can you provide a reference here? I do think your empirical evidence is pretty strong, so you would not necessarily need to include an additional theoretical modelling approach for your selection, but the statement here is somehow isolated. The link to theory could be a bit stronger.

Reviewers' comments:

Reviewer #1 (Remarks to the Author):

The manuscript by Grossen et al. examines patterns of deleterious mutations in six ibex species. They find several interesting patterns, including more putatively weakly deleterious and nearly neutral alleles and homozygous genotypes in more bottlenecked populations. However, strongly deleterious mutations appear to be reduced in the more bottlenecked populations, consistent with the effects of purging of deleterious mutations. This study combines whole genome sequence data with detailed population genetic modeling. As such, it takes a comprehensive approach to addressing important questions in an intriguing study system.

I have a number of major comments with some of the technical aspects of the manuscript:

1) The authors examine 6 populations. This is indeed a strength of the paper. However, the number of genomes sampled from each population varies considerably. For example, 29 Alpine ibex were sampled, while only 4 Iberian ibex were sampled. Some of the analyses presented here are sensitive to sample size and, as such, it is hard to draw meaningful conclusions with such disparate sample sizes. The authors should downsample the populations with a large number of individuals to make things more comparable across populations. Specifically, the results in Figure 1E, the SFSs in Figure 2, and the frequency plots in Figure 5 will be especially sensitive to sample size. It's not really meaningful to compare SFS of different sample sizes to each other.

RESPONSE: We agree that unequal sample sizes may potentially affect the comparisons presented across populations or species in the above-mentioned figures. To remedy this, we performed downsampling and jack-knifing. For Figure 1E, we show now confidence intervals of a downsampling procedure to four individuals (i.e. the sample size of the Iberian ibex) in all species with sample size

above four. For Figure 2, we show now confidence intervals associated with the downsampling of Alpine ibex in the cross-species comparison (updated Figure 2). We have also performed a jack-knife procedure of both Iberian and Alpine ibex to assess the impact of including this specific set of four Iberian ibex individuals (Figure S11). For Figure 5, we similarly performed downsampling of all populations/species to an equal number. Hence, we are able to show now confidence intervals.

In all re-analyses for Fig. 1E, 2 and 5, the downsampling and jack-knife procedures confirmed the previously reported differences in deleterious mutations, excess of rare high-impact variants in Alpine ibex and allele frequency differences, respectively. We believe that the suggested re-analyses have now considerably strengthened the evidence.

2) SFS comparison in Figure 2: The authors conclude the fact that in the Alpine Ibex, more deleterious mutations are skewed more toward low frequency constitutes evidence of purging. I do not think that this difference in the SFS constitutes evidence of purging. Rather, this skew in the SFS would indicate that the high impact mutations are under more purifying selection than the moderate or modifier mutations. This finding is what one would expect to see for deleterious mutations. It does not imply purging. The fact that this pattern is not seen with the Iberian Ibex could be due to the differences in sample size (see my previous comment). Further, Figure 2 uses the proportional SFS here (ie the proportions of segregating SNPs at different frequency) rather than the absolute SFS (ie the numbers of SNPs at different frequencies). Thus, this figure does not show whether there are more or less high impact SNPs. It's just saying that of the SNPs that are there, they are skewed toward lower frequency. We would expect fewer strongly deleterious mutations to be segregating at all and this would not be picked up in the proportional SFS.

RESPONSE: The reviewer is obviously correct that the SFS presented in Figure 2 cannot be taken as independent evidence for purging. Instead, it should be interpreted as evidence for purifying selection. We have previously interpreted this as “strong selection” and are now more specifically using the term “purifying selection”. The relevant evidence for purging here is the *Rxy* analyses presented in the same figure. The *Rxy* shows that Alpine ibex have indeed less segregating highly deleterious mutations.

We are also grateful for the advice to show the absolute SFS instead of the proportional SFS. This is now included in Figure 2. We also show the outcome of the downsampling on the SFS of Alpine ibex in the same Figure.

3) Comparisons of derived alleles and homozygous genotypes across populations can be sensitive to missing data. How was missing data dealt with here? Are levels of missing data similar across populations?

RESPONSE: It is surely true that differences in missing data rates could affect our analyses. However, we used high-coverage Illumina sequencing data and stringently filtered for a maximum missing data rate of 10% per locus. We analyzed what the effective level of missingness is across populations and found that this varied from 0.03% and 0.07%. Furthermore, we found no correlation between missingness and either number of alleles or homozygosity. Hence, any impact from variation in missingness should be minute.

We nevertheless, performed a re-analysis by including only loci with a 100% genotyping rate. We found no meaningful difference in the outcome of the analyses. All these findings are now mentioned in the genotype filtering and population genetics methods section and shown in the new Supplementary Figures (S43, S44).

4) How were fixed sites dealt with in the counts of homozygous derived genotypes across populations? Specifically, if there was a site where all individuals in a population were homozygous for the derived allele, yet in other population, there were 2 alleles segregating, how would this site be counted? What if there are sites that are fixed for different alleles in different populations? How were these counted?

RESPONSE: For the counts of homozygous derived genotypes across populations, we based ourselves on all sites that are polymorphic among Alpine ibex and Iberian ibex. Hence, if a site was monomorphic in a population but there was polymorphism within one of the two species, we retained the site for all population comparisons. We mention this now more explicitly in the population genetics methods section. As mentioned previously, sites were also filtered to be biallelic and retained only if the derived state could be ascertained.

5) Lines 137-144 (and associated figures): The authors compare the numbers of homozygous deleterious mutations and derived alleles across different populations. Are these differences statistically significant? Some sort of statistical test should be presented here. Similarly, in the R_{xy} plots (e.g. Figure 2E), what are the distributions over? How was statistical significance assessed here?

RESPONSE: We performed now statistical tests for data presented in Figures 2 that previously simply reported differences in counts. We found that the numbers of homozygous highly deleterious mutations ($p = 0.015$; Figure 2F) and derived alleles ($p = 0.003$) differed significantly between Alpine and Iberian ibex. For R_{xy} analyses (Figure 2E), we previously showed confidence intervals for jack-knife procedures over chromosomes (following Do et al. 2015, *Nature Genetics*). We now also performed all-against-all tests between all categories (using Tukey HSD). All comparisons were significant ($p < 0.0001$), except the comparison of moderate to low impact mutations ($p = 0.06$). We mention all these analyses now in the main text.

6) Figure 6 and forward simulations: I like how the authors include forward simulations with realistic demographic parameters. However, on line 251, the authors indicate that “Alpine ibex carry an overall higher mutation burden than related species”. This can be directly computed from the model. Specifically, the mean fitness of each of the populations can be computed given the numbers of segregating deleterious mutations, their selection coefficients, and their dominance coefficients. It would be good to compute this actual genetic load from the simulations and see how it compares. There can be differences between the load computed from simulations vs. what’s seen empirically when counting mutations (e.g. see Pedersen et al. 2017 *Genetics*).

RESPONSE: We appreciate the suggestion about computing directly the burden from the model, however line 251 was meant as being part of the conclusions bringing together both the empirical evidence and the outcome of the simulations.

Nevertheless, we pursued the suggestion and investigated species differences in terms of genetic load. Our simulations did not include individuals representing Iberian ibex. The simulated long-term population size of Alpine ibex (pre-species bottleneck) was set at 1000 individuals. This does match the bottleneck size experienced by Iberian ibex.

We computed genetic load defined as the mean realized fitness ($L=1 - W_{\text{mean}}/W_{\text{max}}$, where W_{max} is the maximum number of surviving offspring per female computed from her deleterious mutations). We found an increase in genetic load in Alpine ibex following the species bottleneck (Fig. S17). Hence, the accumulation of mildly deleterious mutations was reducing overall fitness despite purging.

We now briefly mention the outcome of the above analyses in the simulation results and methods paragraphs and show the outcome in Supplementary Figure S17.

7) Simulations: Do the simulated Alpi Maritime have fewer homozygotes for high impact mutations than do the gp? It’s hard to tell from Figure 6c. This finding is important for demonstrating purging.

RESPONSE: We found that the median number of homozygote counts for high impact mutations is lower for Alpi Maritime than Gran Paradiso. However, the difference was not statistically significant (Mann-Whitney U, $p = 0.39$). Evidently, when analysing the outcome of simulations, the number of replicates has a direct impact on levels of significance.

We believe that the more relevant measures here are the allele counts (Simons and Sella Curr Opin Genet Dev 2016) and the R_{xy} (Do et al. Nature Genetic 2015). Both take into account deleterious variants even if completely removed from a specific species or population. R_{xy} provides standardized metrics using intergenic sites making R_{xy} robust against sampling effects and population substructure. We found that allele counts were significantly lower in Alpi Maritime compared to Gran Paradiso ($p = 0.001$). We added now a new R_{xy} analysis comparing Alpi Maritime and Gran Paradiso. We find that high impact mutations are indeed relatively less frequent in Alpi Maritime than Gran Paradiso (Figure S22).

This new information is now mentioned in the relevant results section for the simulations.

8) It would be good to better related Figure 6B-D to the empirical patterns. Right now it's hard to see which simulated pattern goes with which empirical pattern.

RESPONSE: We completely agree. We have now added some additional explanations in the simulations sections comparing the empirical and simulation datasets.

9) Figure S13: Here when GERP scores are used to predict which mutations are likely to be deleterious, in the 6+ category, it does not appear that "am" have fewer homozygous genotypes or derived alleles. Based on this figure there does not appear to be any evidence of purging (ie the more inbred or bottlenecked population having fewer deleterious alleles). Some reasons for this discrepancy with Figure 4 in the main text should be presented. One possibility is that GERP is not doing a good job at distinguishing strongly deleterious mutations from more moderately deleterious ones. I think this needs to be highlighted and discussed more in the main text. This also motivates the need to consider examining mean fitness directly from the simulations, in addition to relying on bioinformatics predictions of fitness effects on empirical data.

RESPONSE: GERP scores define the level of conservation of a chromosomal region across species. While this is clearly an indication that purifying selection is likely to act on this region, synonymous or intergenic mutations in regions of high GERP scores are unlikely to experience strong selection. As the reviewer suggested, GERP scores may indeed be poorer indicators of fitness consequences than a categorization into stop-codon/non-synonymous/synonymous/modifier mutations.

As requested above, we more deeply analyzed our simulation dataset to show how realistic bottleneck strengths can indeed lead to purging (and how fitness may be affected). In addition, we also present a variety of complementary statistics that are designed to assess the fitness consequences of individual mutations (please see our response to Reviewer 2).

It is also important to note that the only other existing evidence for purging in a wild species was also using annotation-based impact information rather than GERP scores (Xue et al. 2015, Science).

We have modified the sentences in the results describing the outcome of the GERP vs. SnpEff scoring. We also mention now that it would be relevant to extract mean fitness from the simulations as suggested. In response to Reviewer #2, we have now also performed four additional scoring methods that do only to a minor degree rely on conservation (see below).

Additionally, I have several minor suggestions to improve the paper:

1) Lines 55-59: There is some conflating between the effects of genetic drift allowing weakly deleterious mutations to increase in frequency and the effects of inbreeding (ie mating between close relatives) throughout the manuscript. Though they can often occur together, especially in small populations, these are distinct processes, and can have different effects on weakly deleterious vs. strongly deleterious mutations as well as additive vs. recessive mutations. Thus, I would urge the authors to use the terminology more carefully and precisely throughout the paper. Also, line 162 is problematic for this same reason.

RESPONSE: We agree that multiple sentences lacked clarity and have now revised these in light of the above comment.

2) Line 65: Some additional references to cite here include: Do et al. 2015 Nature Genetics; Lohmueller 2014 Curr Opin Genet Dev; Fu et al 2014 AJHG.

RESPONSE: We thank the reviewer and have now included the references.

3) Figure 5: I like this figure for showing the locations of the highly deleterious mutations in the proteins. However, I'm not sure what I'm supposed to conclude from the panel at the right showing the frequencies of these mutations in the different populations. More context and interpretation should be provided.

RESPONSE: We added the allele frequencies across species/populations purely to allow readers to investigate whether a mutation in a particular gene was frequent in any given population/species. We did not intend this as a test of any sort.

4) Lines 257-259: Here the authors seem to imply that bottlenecks may be good because they allow for purging of deleterious mutations. However, this is conditional on the population surviving and not going extinct due to inbreeding depression in the meantime. The Isle Royale wolves offer a counter example. Here, small population size combined with strong inbreeding led to severe inbreeding depression. Robinson et al 2019 (Science Advances) provides a description and analysis of this system.

RESPONSE: We were not sufficiently clear in our wording. A few lines further up we wrote: "Importantly, purging was effective against highly deleterious mutations. Mildly deleterious mutations actually accumulated through the reintroductions. Hence, the overall number of deleterious mutations increased with bottleneck strength"

We tried to clarify the text and now cite also Robinson et al. (2019) as an example for a failure in purging likely due to an overly extreme bottleneck.

5) Lines 309: The authors detect ROHs using BCFtools. As this method relies on allele frequencies, I would suggest that it would be good to also compare to the ROHs seen using other programs, like PLINK. (I do agree that BCFtools is the preferred approach, but as a sanity check, it would be good to see whether the broad qualitative trends hold with PLINK too).

RESPONSE: We have now reanalyzed evidence for ROH using the alternative approach with PLINK. We found that the qualitative outcome was not affected by the method to estimate ROHs. We mention this sanity check now in the relevant methods section. The figures are now shown as supplementaries (Figures S41 and S42).

6) Line 396: Is this initial frequency of the deleterious mutations at mutation-selection balance used at the start of the burn in?

RESPONSE: Yes, this is now clarified in the relevant methods section.

7) Figure S6: The GERP score thresholds presented here are a bit confusing. The figure says “only retain high quality deleterious mutations: GERP>2”. Many mutations with GERP>2 won’t be deleterious. Also, a few lines down in this figure, it would be better to order the GERP and the snpEFF functional predictions in the same order (e.g. low to high for both).

RESPONSE: This was indeed not clearly phrased and presented. We now write “... high-quality mutations detected in conserved regions” and have rearranged the GERP score categories in the concerned figure.

Reviewer #2 (Remarks to the Author):

The authors sequenced and analyzed 60 complete genomes of six ibex species and the domestic goat to investigate the relation between bottleneck strength and mutation load. These ibex species experienced the most dramatic bottlenecks among the successfully restored species. For instance, Alpine ibex were reduced to ~100 individuals in the 19th century and now about 50'000 individuals have been re-established in Alps. They concluded that historic bottlenecks, temporary reductions in population size, induced by human activity impacts levels of genome-wide variation and caused both relaxed selection and purging that reshaped landscape of deleterious mutation load. This study also proposed that for ensuring long-term survivability of endangered species, conservation efforts should model bottleneck intensity. In their study, defining the phenotypic effects of variants is an important issue, and the authors defined the mutational effects using snpEff toolbox and GERP conservation scores for genes with evidence for transcription. However, two major problems exist in defining high-confidence deleterious mutations.

Major comments:

1. Four categories proposed by snpEff are HIGH, MODERATE, LOW, and MODIFIER, which is a simple estimation of putative impact/deleteriousness for the Sequence Ontology terms. For instants, snpEff categories stop-gain, frameshift or start-lost variants as high putative impact, and missense variant, in-frame deletion and insertion with moderate impact. As we know, many missense mutations can cause diseases. Generally, the possibility or the fraction of variants with high putative impact to be deleterious mutations should be higher than moderate or low impact mutations. However, the sequencing results show that the number of high impact mutations by snpEff is much less than other categories. In Table S4, the number of SNPs is 614 and 70,873 classified as high and moderate respectively for all ibex species.

RESPONSE: We appreciate these requests for clarifications and re-analyses. It is true that among high-impact mutations, there is a higher probability for deleterious effects in contrast to e.g. low-impact mutations. It is also true that as selection has already acted on the polymorphism of any species under consideration, high-impact mutations tend to be rather rare despite the fact that numerous mutational targets would exist in the genome. E.g. one would obviously never observe any dominant lethal mutations. Consistently with this, we observed 614 high-impact (i.e. frameshift/start/stop mutations) and 70,873 moderate (non-synonymous mutations), as the former category should be much less tolerated than the latter category.

We were not clear enough when referring to Table S4. The table reports the total number of observed intergenic SNPs passing our stringent filtering. Following the above, there is an expectation that low-impact or modifier mutations should be more abundant than high-impact mutations. This is partly due to the fact that there is less opportunity for mutations to lead to high-impact as for instance stop-gain. High-impact mutations are furthermore expected to be relatively underrepresented in most populations because purifying selection tends to remove these mutations while low-impact mutations should be relatively unaffected. This should explain the numbers observed in Table S4.

Even though 1% of moderate variants suggested by snpEff have the possibility to be deleterious mutations, the number of it is about 708. Therefore, the authors should provide a more rigorous definition for the phenotypic effects of variants. For instance, using several other popular prediction methods (such as Revel, Vest or CADD) to find the deleterious mutations in moderate category and then do further analysis.

RESPONSE: We are fully in agreement that a more rigorous understanding of fitness consequences of individual mutations requires independent analyses using different scoring approaches. We have now implemented the three above-mentioned prediction methods plus a fourth popular tool (SIFT). These tools allow us to perform more detailed analysis for the moderate category (and others) as requested. We used the scores assigned to individual mutations to confirm the two major findings in our study: the *Rxy* analyses showing purging of highly deleterious mutations in Alpine ibex versus Iberian ibex, as well as the *Rxy* analysis showing purging in the most severely bottlenecked Alpine ibex population from Alpi Marittime compared to other Alpine ibex. We have summarized all the outcomes in a new figure (new Figure S12).

As we apply these tools outside of humans, we obviously do not have pathological evidence underpinning score cut-offs for deleteriousness. We conservatively used score cut-offs proposed as best-practices by the tool developers or used a very conservative percentile (i.e. VEST3). Details about each individual analyses and choice of cut-offs are now added to the methods (population genetics section) and the outcome mentioned in the results.

Overall, our extensive validation of our findings using different assessments of levels of deleteriousness showed that REVEL, SIFT, VEST3 and CADD confirm our findings of purging in Alpine ibex compared to the Iberian ibex. The evidence of purging for the most severely bottlenecked Alpine ibex population from Alpi Marittime compared to other Alpine ibex was also consistently supported by REVEL, SIFT, CADD and VEST3 classifications.

2. The authors assessed the severity of a variant using GERP conservation score. There are another two popular conservation scores, phyloP and phastCons, and I suggest the authors to validate their conclusion using one of these scores.

RESPONSE: We appreciate these important suggestions and have now included both phyloP and phastCons. We used best-practice to define a cut-off distinguishing deleterious from neutral mutations (see methods). Then, we repeated the key site frequency spectrum (SFS) analysis based on GERP conservation scores each with the two different conservation scores. In all cases, we can now show that Alpine ibex experienced purifying selection of deleterious mutations as indicated previously by GERP scores alone. We mention these additional tests now when discussing Figure 2B/D and present the results in a new supplementary figure (Figure S10).

Minor

comments:

1. I cannot find GERP scores from <http://mendel.stanford.edu/SidowLab> that the authors provided.

RESPONSE: The link seems to have been modified. We mention now the updated link (<http://mendel.stanford.edu/SidowLab/downloads/gerp/>) in the methods section (Identification of high-confidence deleterious mutations, second paragraph).

2. Please illustrate why the authors choose the cutoffs of GERP > -2 and the minimal distance to the next SNP of 3bp.

RESPONSE: We regret that the justifications were not made clear enough. Our aim was to retrieve categories of neutral to highly deleterious mutations. Population genetics analyses in human populations defined a cut-off of +2 and above to identify moderate to highly deleterious mutations (e.g. Henn et al. PNAS 2015). As in previous work, we considered mutations -2 to +2 to be neutral (or at least very close to neutral). Mutations with GERP < -2 were not considered in order to avoid regions of very low conservation and potentially challenging to properly contrast among species.

The minimal distance of 3 bp between adjacent SNPs was introduced to avoid biases due to multi-nucleotide polymorphisms (MNPs). We considered loci with potential MNPs to be problematic to assess across populations/species and how selection impacted these sites. We preferred to conservatively exclude any SNP at less than 3 bp to another.

We have added more explanations to the relevant method sections (Identification of high-confidence deleterious mutations, last paragraph).

Reviewer #3 (Remarks to the Author):

Authors here report a genomic study on the Alpine Ibex in the Alps. This species can be seen as a model species for investigating the role of bottlenecks on demography as well as shaping genomic structure. Thus the study set provides a unique opportunity for investigating the role of purging under in natural populations when the species has been facing strong bottlenecks. So far only very few studies have actually been undertaken trying to identify the role of selection and estimating the mutation load in natural systems and the current study does fills an important gap. The strength of the manuscript is the combination of a well documented demographic history and thus the combination of population sizes, bottleneck history and the genomic analyses. Relating purging to the different history and population scenarios definitely is highly relevant for a conservation perspective. Analyses are novel and appropriate and the appendix provides a massive amount of background data. Thus I do think the paper will make a significant contribution not only to the field of Conservation Biology and Genetics but also to understand evolutionary processes in the wild.

I have two points, I think, authors should address, though:

Sample size and distribution is not equal for all investigated populations. While I understand that the populations in the Alps have been in the focus also of previous studies and thus most material comes from this population I wonder how representative the sampled genomes of the other species are. Some have only two genomes and probably are opportunistically be sampled, but as information is missing on population sizes of the other Ibex species (Table S1A and TableS1B are only for Alps and Iberian Ibex) it is a bit unclear to me, if those few samples from the other populations are good representatives and if and how authors have dealt with that fact. I would also recommend to add not only rough population sizes for the other species, but also to include sampling dates for all genomes in the appendix.

RESPONSE: We fully agree that the potential impact of unequal sample sizes, in particular for Alpine ibex and Iberian ibex, required further investigation. Reviewer #1 has raised the same issue and we have in response made a number of new analyses (kindly see our responses above). In particular, we have performed downsampling and jack-knifing to understand whether the conclusions were robust.

The figures 1, 2 and 5 have now been revised to show confidence intervals from these resampling procedures. In summary, our conclusions were well supported.

In addition, we have now added all known sampling dates for all genomes and population sizes for each species at the first mention in the main text.

line32: sentence reads as if something is missing here

RESPONSE: Improved.

line46/47: at several places I found the use of tenses inconsistently, as e.g. in the first and second sentence of the introduction – please check.

RESPONSE: We have now carefully checked this and hope that all is consistent now.

line 55: “purging.Purging” more a stylistic remark, but could be reworded

RESPONSE: Improved.

Figure 1: line 78: in addition to the IUCN conservation status it would be good to add rough population sizes in the boxes for each species as well

RESPONSE: We added population sizes as suggested. This is mentioned in the main text when introducing the species for the first time.

line 80/87: how do sample sizes match – in the figure N=29 is mentioned for Alpine Ibex, in the text you refer to 60 genomes (which is probably rather a matter of wording, but I first was confusing that with 60 Alpine Ibex samples)

RESPONSE: We clarified that we sequenced 29 Alpine ibex genomes, which were part of the 60 genomes analyzed in total.

line116/117: So what is the explanation for this? I did not find that afterwards.

RESPONSE: We clarified that this is consistent with purifying selection acting more efficiently against highly deleterious mutations in Alpine ibex compared to Iberian ibex.

line120: I can understand the rationale of contrasting Iberian Ibex to Alpine Ibex in that graph, however, I would like to see a comparison to the other species (probably in appendix and domestic goat species as well)

RESPONSE: We agree that it would be very interesting to more broadly compare site frequency spectra (SFS) and R_{xy} across species. However, to do this would require proper sampling of representative individuals from all other ibex. We are afraid that performing SFS and R_{xy} on 1-2 individuals would be impossible or inappropriate. For the domestic goat, a suitable population would also have to be

identified. We have included though summary statistics in Figure 1E for deleterious mutations observed across species. We hope that this overview serves as a general representation of the level of deleterious mutations across species.

line143/144: is this true also when you analyse the different alpine ibex populations separately?

RESPONSE: We have indeed performed analyses on the evolution of mildly deleterious versus highly deleterious mutations among different Alpine ibex populations. We present these analyses in Figures 3 and 4. We found that the most heavily bottlenecked population of Alpine ibex accumulated mildly deleterious mutations but purged heavily deleterious mutations (Alpi Marittime; Figure 3). Homozygote and allele counts across populations are shown in Figure 4. We have generally revised the text describing these analyses and we hope that the new version is more satisfactory.

line259/261: can you provide a reference here? I do think your empirical evidence is pretty strong, so you would not necessarily need to include an additional theoretical modelling approach for your selection, but the statement here is somehow isolated. The link to theory could be a bit stronger.

RESPONSE: We tried to improve the link between the empirical data and modelling in the paragraph describing the simulations. We added a reference to Lynch et al. 1995 (Am Nat).

Reviewers' Comments:

Reviewer #1:

Remarks to the Author:

Overall the authors were very responsive to my comments on the previous version of the manuscript. I think the new manuscript is considerably stronger and more convincing than the first version.

I have a number of remaining comments to improve the paper:

1) Lines 121-125: I applaud the authors for doing the SFS analyses on similar sample sizes across populations. However, I think the results in Figure 2 are hard to directly interpret. It appears that the high effect mutations in the Alpine Ibex are skewed toward lower frequency, conditional on segregating. This itself is an interesting result, but I don't think it by itself shows that "purifying selection acting more efficiently against highly deleterious mutations in Alpine ibex compared to Iberian ibex". For example, there could just be fewer high impact mutations in the Iberian Ibex overall. I think the R_{xy} analysis (Figure 2E) better makes the point the authors are trying to make about strongly deleterious mutations being more effectively selected out in Alpine Ibex.

2) Figure 2B: Indicate what the different colors mean.

3) Line 183: It would be good to discuss Figure 3E (the R_{xy} analysis) here before talking about Figure 4.

4) Lines 210-211: Interestingly, it seems that GERP may do well with distinguishing neutral from selected mutations. However, GERP (and other conservation-based metrics) probably won't do as well at distinguishing weakly deleterious from strongly deleterious mutations because both will not accumulate as fixed differences between species. This may be why explicitly considering mutations to premature stop codons shows evidence of purging (ie they are enriched for strongly deleterious mutations) while GERP does not.

5) I still find Figure 5 to be a bit misleading. In the plot with the frequencies on the right, it seems like most of these mutations are absent from the Iberian Ibex. One may naively think that this implies more deleterious mutations in Alpine Ibex. However, these mutations in Figure 5 were ascertained as being found in Alpine Ibex. As such, it's not surprising that they are at higher frequency in Alpine Ibex compared to Iberian. One solution could be to directly state this in the legend of Figure 5 to avoid the readers making improper comparisons.

6) Lines 300-301: I think the authors do a good job at contrasting the results between weakly deleterious and strongly deleterious mutations. They correctly point out that the weakly deleterious can accumulate. However, I think the importance of weakly deleterious vs. strongly deleterious mutations for short-term extinction risk is a bit unclear. In other words, while the more weakly deleterious mutations are accumulating, they may not quickly lead to extinction. The Isle Royale wolf example demonstrate this contrast nicely. So, I would recommend that the authors discuss this subtlety a bit more at the end of the paper.

7) Line 63: Close the ")".

8) Line 473: Please indicate that this gamma distribution on s only assumes deleterious mutations.

9) Figure S14: What do the numbers at the top represent? "1" "1-2", etc?

10) Line 481: I'm not sure a burn in of 3000 generations is long enough to reach mutation-drift-selection equilibrium for a population of size 1000. It would be good to run the burn in longer and show whether summary statistics of diversity look like those in the 3000 burn-in simulations.

Reviewer #2:

Remarks to the Author:

The authors have addressed all my comments.

Reviewer #3:

Remarks to the Author:

Authors have been addressing all comments raised by reviewers carefully and I feel that the modifications/explanations now helped to create a more concise version of the manuscript.

I only have a few minor points:

line 74/75: in the abstract you always refer to bottlenecks (plural) and in this paragraph you also indicate the one strong bottleneck in the 19th century (line 72) and then adding the following sentence "most extant populations...." While anyone familiar with the study system know that each mountain ridge or valley might have experienced additional bottlenecks the history is not clear to everyone. I would add another half sentence here explaining that there have been many local colonies established – mixing the whole Alpine ibex population with local colonies and their demography needs to be somehow explained – I do think this does not need comprehensive but could be done with a half additional sentence and rephrasing – especially as you bring some information in lines 164 onwards line 94 is redundant to line 70

REVIEWERS' COMMENTS:

Reviewer #1 (Remarks to the Author):

Overall the authors were very responsive to my comments on the previous version of the manuscript. I think the new manuscript is considerably stronger and more convincing than the first version.

I have a number of remaining comments to improve the paper:

1) Lines 121-125: I applaud the authors for doing the SFS analyses on similar sample sizes across populations. However, I think the results in Figure 2 are hard to directly interpret. It appears that the high effect mutations in the Alpine Ibex are skewed toward lower frequency, conditional on segregating. This itself is an interesting result, but I don't think it by itself shows that "purifying selection acting more efficiently against highly deleterious mutations in Alpine ibex compared to Iberian ibex". For example, there could just be fewer high impact mutations in the Iberian Ibex overall. I think the Rxy analysis (Figure 2E) better makes the point the authors are trying to make about strongly deleterious mutations being more effectively selected out in Alpine Ibex.

Response: We have already adjusted the interpretation of the SFS in the previous revision. The reviewer suggests that we claim "shows that purifying selection acting...". We wrote however "This is consistent with purifying selection acting...". The reviewer has previously agreed with us that the signature was indeed consistent with purifying selection but no demonstration of it. Hence, we believe that our formulation already satisfied the reviewer's request. We write then further below, as suggested previously as well, that Rxy is the actual demonstration of purifying selection against strongly deleterious mutations.

2) Figure 2B: Indicate what the different colors mean.

Response: The missing information has been added to Figure 2B.

3) Line 183: It would be good to discuss Figure 3E (the Rxy analysis) here before talking about Figure 4.

Response: We mention now the Rxy analysis in the more appropriate location as suggested.

4) Lines 210-211: Interestingly, it seems that GERP may do well with distinguishing neutral from selected mutations. However, GERP (and other conservation-based metrics) probably won't do as well at distinguishing weakly deleterious from strongly deleterious mutations because both will not accumulate as fixed differences between species. This may be why explicitly considering mutations to premature stop codons shows evidence of purging (ie they are enriched for strongly deleterious mutations) while GERP does not.

Response: This is indeed an interesting observation and we added now a sentence about this.

5) I still find Figure 5 to be a bit misleading. In the plot with the frequencies on the right, it seems like most of these mutations are absent from the Iberian Ibex. One may naively think that this implies more deleterious mutations in Alpine Ibex. However, these mutations in Figure 5 were ascertained as being found in Alpine Ibex. As such, it's not surprising that they are at higher frequency in Alpine Ibex compared to Iberian. One solution could be to directly state this in the legend of Figure 5 to avoid the readers making improper comparisons.

Response: We added this information to the legend as suggested.

6) Lines 300-301: I think the authors do a good job at contrasting the results between weakly deleterious and strongly deleterious mutations. They correctly point out that the weakly deleterious can accumulate. However, I think the importance of weakly deleterious vs. strongly deleterious mutations for short-term extinction risk is a bit unclear. In other words, while the more weakly deleterious mutations are accumulating, they may not quickly lead to extinction. The Isle Royale wolf example demonstrate this contrast nicely. So, I would recommend that the authors discuss this subtlety a bit more at the end of the paper.

Response: We clarified now the context of mutation accumulation and extinction risk in the final paragraph.

7) Line 63: Close the ")".

Response: Fixed.

8) Line 473: Please indicate that this gamma distribution on s only assumes deleterious mutations.

Response: The information has been added.

9) Figure S14: What do the numbers at the top represent? "1" "1-2", etc?

Response: This refers to the number of bottlenecks. This is now clarified in the figure legend.

10) Line 481: I'm not sure a burn in of 3000 generations is long enough to reach mutation-drift-selection equilibrium for a population of size 1000. It would be good to run the burn in longer and show whether summary statistics of diversity look like those in the 3000 burn-in simulations.

Response: We chose a burn-in of 3000 generations based on two important parameters: the number of segregating deleterious mutations and heterozygosity at deleterious mutation loci. As suspected by the reviewer, a burn-in of 3000 generations is indeed not long enough to reach mutation-drift-selection equilibrium. However, at equilibrium all deleterious mutations would be fixed as both population size and number of loci are finite. This would defeat the purpose of simulating the evolution of deleterious mutation load. The number of segregating deleterious mutations and heterozygosity at deleterious mutation loci changed rapidly early in the burn-in and reached a plateau at ~3000 generations before slowly decreasing. We clarified all this in the methods.

Reviewer #2 (Remarks to the Author):

The authors have addressed all my comments.

Reviewer #3 (Remarks to the Author):

Authors have been addressing all comments raised by reviewers carefully and I feel that the modifications/explanations now helped to create a more concise version of the manuscript.

I only have a few minor points:

line 74/75: in the abstract you always refer to bottlenecks (plural) and in this paragraph you also indicate the one strong bottleneck in the 19th century (line 72) and then adding the following sentence "most extant populations..." While anyone familiar with the study system know that each mountain ridge or valley might have experienced additional bottlenecks the history is not clear to everyone. I would add another half sentence here explaining that there have been many local colonies established – mixing the whole Alpine ibex population with local colonies and their demography needs to be somehow explained – I do think this does not need comprehensive but could be done with a half additional sentence and rephrasing – especially as you bring some information in lines 164 onwards

Response: Expanded sentences slightly.

line 94 is redundant to line 70

Response: We rephrased the latter sentence.